

# Characterising the behaviour of surge and non-surge type glaciers in the Kingata Mountains, Eastern Pamir, from 1999 to 2016

LV Mingyang[1,2,4] GUO Huadong[1,2] LU Xiancai[1] LIU Guang[2*] YAN Shiyong[3] RUAN Zhixing[2] DING Yixing[2] Duncan J. Quincey[4]

[1]School of Earth Sciences and Engineering, Nanjing University, Nanjing, 210023, China

[2]Key Laboratory of Digital Earth Science, Institute of Remote Sensing and Digital Earth, Chinese Academy of Sciences, Beijing, 100094, China

[3]Jiangsu Key Laboratory of Resources and Environmental Engineering, School of Environment Science and Spatial Informatics, China University of Mining and Technology, Xuzhou, 221116, China

[4]School of Geography, University of Leeds, Leeds, LS2 9JT, UK

*Correspondence to:* LIU Guang(liug@ceode.ac.cn)

**Abstract.** Glaciers in the Pamir Mountains are generally acknowledged to be in a stable state and show the least glacial retreat in High Mountain Asia; however, they are also some of the most dynamic glaciers in the region and their behaviour has been spatially variable in recent decades. Few data exist for these glaciers, in particular relating to how they are responding to recent climatic changes. Here, we utilize Landsat 7, Landsat 8, and ASTER optical images acquired between 1999 and 2016 to characterise the dynamics of the glaciers in the Kingata Mountains, located in the eastern Pamir. We quantify the velocity, areal, and frontal changes of these glaciers, which provide us with valuable data on their recent dynamic evolution and an indication of how they may evolve in future years. We highlight 28 glaciers among which 17 have changed markedly over the study period. We identify 4 advancing glaciers, 1 receding glacier, and 12 surge-type glaciers. The dynamic evolution of the glacier surges shows some similarity with those of the nearby Karakoram, suggesting that both hydrological and thermal controls are important for surge initiation and recession. Topography seems to be a dominant control on non-surge glacier behaviour in the KM, with the north-side of the divide characterised by steep, avalanche-fed basins and glacier tongues now approaching recession in contrast to those on the south-side of the divide that capture the majority of precipitation and have

much broader plateau-like accumulation zones. This study is the first synthesis of glacial motion across this region and provides

a baseline with which to compare future changes.

## 1 Introduction

Changes in mountain glaciers represent a key indicator of climate variability, as well as contributing to sea-level-rise and at

5   times being the source of flood hazards (Oerlemans, 1994; Meier et al., 2007; Gardner et al., 2013; Burgess et al., 2013). High

Mountain Asia (HMA), extending from the Hindu Kush and Tien Shan in the west to the glaciers and snowfields of Yunnan,

China, in the east, contains the greatest mass of ice outside the Earth's polar regions. Since the 1990s, the glaciers in HMA

have broadly been in recession, and the glacier mass balance has been negative (Yao et al., 2004; Tian et al., 2006). Rates of

recession generally decrease towards the western part of the region (Yao et al., 2012). Indeed, in the far west, the eastern Pamir

is characterized by relatively stable glaciers and, although there remains debate within the literature regarding the details

(Khromova et al., 2006; Bolch et al., 2012; Kääb et al., 2012; Farinotti et al., 2015), recent research has revealed that Pamir is

the region of transition from positive to negative mass balance, with the eastern Pamir experiencing a slight positive mass

balance (Gardner et al., 2013; Gardelle et al., 2013; Osmonov et al., 2013; Holzer et al., 2015; Kääb et al., 2015; Brun et al.,

2017).

Within the eastern Pamir lies the Kingata Mountains (KM) (Fig. 1). Due to their remoteness and harsh environment, few

ground-based glaciological studies have been carried out (Thompson et al., 1995; Ono et al., 1997; Shangguan et al., 2016).

Remote-sensing studies have provided most of the available data for this region, using both optical and radar imagery to

evaluate the temporal changes in glacier extent and flow (Kääb, 2002; Luckman et al., 2007; Yang et al., 2013; Yasuda and

Furuya, 2015). Three approaches have generally been employed to derive displacements of the glacier surface: offset tracking

based on intensity using both optical and synthetic aperture radar (SAR) imagery (Copland et al., 2009; Quincey et al., 2011;

Burgess et al., 2013; Quincey et al., 2015), interferometric SAR (InSAR) (Gourmelen et al., 2011; Kenyi and Kaufmann, 2003;

Goldstein et al., 1993), and a combination of these two methods (Yan et al., 2016). Images acquired by optical sensors are

often limited by cloud- and snow-cover as well as coarse spatial resolutions, but the existing global archive offers many possibilities to match features across image pairs over annual to multi-annual timescales. Cross-correlation provides a quick and efficient way of measuring glacier surface displacement and is routinely applied to images acquired by the Advanced Spaceborne Thermal Emission and Reflection Radiometer (ASTER) and Landsat sensors (Luckman et al., 2007; Scherler et

al., 2008; Quincey et al., 2015).

Glacier surges represent an endmember of glacier flow, and remain relatively poorly understood despite an increase in related studies in recent years (Gladstone et al., 2014; Quincey et al., 2015; Sevestre and Benn, 2015; Yasuda and Furuya, 2015). Most surges occur in regular cycles consisting of active phases and quiescent phases (Meier and Post, 1969). During the active phase of a surge, glacier velocities can increase by one or two orders of magnitude within a few months, often accompanied

by an advance of the glacier tongue as large volumes of ice are discharged from high-to-low elevations (Harrison and Post, 2003; Clarke, 1987; Raymond, 1987). During quiescent phases, these glaciers recharge their ice reservoirs at high elevation and recede across their ablation area. Since glacier surges are closely related to glacial hazards such as glacial lake outburst floods (GLOFs) and debris flows, research on their controls and evolution has become a particular focus within the literature (Kargel et al., 2005; Gladstone et al., 2014; Dunse et al., 2015). Glacier surges are reported all over the world and HMA,

particularly the Pamirs, Karakoram, and Tien Shan, is regarded as one of the most active surge zones (Sevestre and Benn, 2015). An inventory of glacier surges for Pamir Plateau in Tajikistan based on remote sensing revealed that 51 out of 215 glaciers had signs of dynamic instability and were considered as surge type (Kotlyakov et al., 2008). However, this inventory did not include the surges in the Pamir in China or investigate any changes in glacier dynamics that have occurred over recent years.

The objective of this paper is, therefore, to characterize the dynamics of glaciers within the KM. In doing this, we will also be able to identify the distribution and characteristics of surge-type glaciers in this region and describe their evolution for the first time and make comparisons with studies from neighbouring regions. Supported by analyses of glacier terminus position and areal change, we aim to quantify the behaviour of glaciers within this region over a seventeen-year period. In doing so, this

study will provide valuable baseline data for future studies focussing on the glaciers of this region, and will help stakeholders

such as regional water resource managers to better understand the likely evolution of these important water sources in coming

decades.

## 2 Study Area

The KM, located on the eastern Pamir Plateau (38°N–39°N, 73°E–75°E), together with the neighbouring Kongur Mountains

and Muztag Ata, form the main mountain range in the eastern Pamir. The mountain range is oriented approximately southeast–

northwest, extending for a length of 120 km and width of 40 km (Fig. 1). The highest summit of the KM is Chakragil, 6760 m

above sea level (asl). Two other major peaks exist in the middle and northern parts of the range, Buduk Seltau (6110 m asl)

and Aksay Bax (6102 m asl). In the southwest, the Muji River flows along the southern slopes of KM towards the southeast.

In the north of the south-eastern end of the mountain range lies the Oytag Glacier Park.

Since the study area is one of the coldest and driest glacierized areas in middle latitude regions, the glaciers are considered to

be continental in type (Zhang, 1980; Su et al., 1989; Li et al., 2004; Jiang et al., 2014). They provide a critical natural water

reservoir for people living downstream (Yao et al., 2004) and changes in glacier extent or behaviour can impact directly on

their ability to be able to irrigate their land and daily water use. Based on data from Taxkogan meteorological station (37°46'N,

75°14'E; 3090.9 m asl), annual precipitation is less than 70 mm. Peak ablation generally occurs between June and August,

when the mean temperature is as high as 15 °C and the monthly mean air temperature over the glacier termini can be higher

than 0 °C (Shangguan et al., 2006). The climate is largely controlled by high-altitude westerly circulation with precipitation

sourced at middle latitudes from the Mediterranean, Black Sea, and Caspian Sea (Su et al., 1989; Seong et al., 2009; Jiang et

al., 2014).

According to the Second Glacier Inventory Dataset of China, there are more than 216 glaciers in the KM, with a total area of

551.16 km$^2$ (Guo et al., 2014). Their terminus elevations vary from 3100 to 5200 m asl, and their mean slopes range from 20°

to 40°. Glaciers on the southern side of the mountain range are generally smaller than those on the northern side, and most

glaciers on the northern side are heavily debris-covered, which is likely to impact their dynamic evolution as well as long-term ablation rates (Scherler et al., 2011; Huang et al., 2018). Many glaciers in this region are small in size (area of less than 2 km$^2$); here we focus on 28 glaciers with area greater than 5 km$^2$ as they provide the greatest contribution to downstream river discharge and remote sensing observations can be made with high accuracy.

## 3 Data and Methods

### 3.1 Satellite Imagery and DEM

We obtained 13 Landsat 7 Enhanced Thematic Mapper Plus (ETM+) images from 1999 to 2013, 17 Landsat 8 Operational Land Imager (OLI) images from 2013 to 2017, and 12 ASTER L1T images from 2003 to 2013, with minimal cloud and snow cover. Imagery from late summer and autumn (~July to October) was preferentially selected to minimize snow cover (Table 1). Several Landsat 2 and Landsat 5 images were also obtained for the identification of surge-type glaciers and surge cycle periods. These images were acquired from the United States Geological Survey (USGS) Earth Resources Observation and Science Center (EROS) (http://glovis.usgs.gov/). Feature tracking was mainly conducted using the Landsat 7 and Landsat 8 images. However, due to the Landsat 7 ETM+ scan line corrector failure in May 2003, several ASTER images were used for feature tracking from 2003 to 2013. All Landsat 7, Landsat 8, and ASTER data were used for glacial boundary identification. Landsat 7 and Landsat 8 images were preprocessed using the USGS Level-1 Product Generation System, which includes radiometric and geometric correction and transformation to UTM projection. Due to the limited number of cloud-free images available for velocity detection and in order to minimise the effect of seasonal velocity variations on our data, we selected image pairs separated by approximately 365 days for the analysis. All the imagery used in this study is listed in Table S1. The ASTER Global Digital Elevation Model (GDEM Version 2) of this area (N38°-N39°; E074°-E075°) was acquired from NASA's Earth Observing System Data and Information System (EOSDIS) and was used for interrogating glacier surface elevation profiles in support of the velocity analyses.



### 3.2 Determination of glacial motion

Glacier surface displacement was quantified by performing co-registration and correlation of the optical imagery using the software package COSI-Corr (Co-registration of Optically Sensed Images and Correlation), described in detail by Leprince et al. (2007) and Scherler et al. (2008). Images were first co-registered to a common reference dataset such that misalignment was less than a single pixel for any pair. The correlation step then uses a sliding window to measure horizontal displacements along the East/West and North/South directions separately. This technique has been validated on several case examples of horizontal ground displacements (Ayoub et al., 2009; Herman et al., 2011; Avouac et al., 2006) and proved to be an effective tool for deriving glacier surface displacement from optical satellite imagery (Heid and Kääb, 2011). Horizontal displacements were measured with the frequential sub-pixel correlator using a multi-scale approach where the correlation window size changed from $32 \times 32$ to $8 \times 8$, sliding every pixel. Using a UTM projection, the measured column and row displacements were combined to obtain the magnitude and direction of glacier surface displacement. The glacier velocity (*GV*) was acquired using Eq. (1):

$$GV = \frac{D \times A_y}{A_i} \tag{1}$$

where $D$ is the displacement measured between each image pair; $A_i$ is the interval between the image pair in days; and $A_y$ is the number of days in the year. The unit of *GV* is m/yr.

We selected 28 glaciers with areas (including the debris-covered part) of more than 5 km$^2$. These 28 glaciers are all recorded in the Second Glacier Inventory Dataset. Several of them are composite glaciers with multiple tributaries feeding a main glacier trunk. The annual velocity fields of these glaciers were derived from 1999 to 2003 utilizing Landsat 7 images, from 2003 to 2011 utilizing ASTER images, and from 2013 to 2016 utilizing Landsat 8 images. For most glaciers, we derived the annual velocity fields in two periods, 1999–2003 and 2013–2016, to highlight any evolution of the glacial motion patterns. In order to complete the time series and investigate motion variability in greater detail, we incorporated ASTER images from 2003 to 2011 to derive the displacements of several surge-type glaciers during the course of their surge events.

Errors in the derived data are reflected by the residual value in the motionless (stable terrain) area. We statistically analysed the residual error in the non-glacier region without rugged topography. The mean uncertainty was 1.6 m/yr with a corresponding standard deviation of 1.3 m/yr. Velocity profiles for different years were extracted along the centreline of each glacier to consider the interannual variability. Due to the heavy debris cover, different snow-cover conditions, and relatively

small size for some parts of the glaciers, decorrelation of the image pair is an unavoidable problem for monitoring glacial motion in this region. In order to reduce the impact of decorrelation, we smoothed the velocity map using Gaussian Low Path filtering method and manually removed clear blunders from the resulting data. This quantitative measurement provides sufficient precision to characterize the glacial motion patterns in the study area.

**3.3 Glacier Delineation and Identification of Surge-type Glaciers**

Glaciers in the KM are heavily covered by debris making the delineation of their boundaries a challenging task. Considering the paucity of glacial research in the study area, we utilized the Second Glacier Inventory Dataset of China (Version 1.0) to identify the glacierised area (Guo et al., 2014). Some glaciers converge together at lower elevations, in which case we treat them as a single ice mass. In order to understand and quantify the evolution of these glaciers, we manually measured the changes in glacier termini for each time period using ArcMap 10.3.1. The accuracy of manual measurement is within 30 m, or

2-pixels for the Landsat panchromatic data of 15 m.
When a glacier experienced some special surface features, such as moraine looping or folding in the middle of the glacier, ice foliation, glacier surface crevassing, or sudden advance of its tongue, we identify the glacier as surge-type (Meier and Post, 1969; Barrand and Murray, 2006). In some cases, discrete changes in glacial motion during a certain time can be taken as the evidence of a surge event.

## 4 Results

Through visual inspection of the satellite imagery we identified 17 glaciers out of 28 studied glaciers that had clearly changed either in terminus positions or in surface features between 1999 and 2016 (Fig. 2). By directly comparing the differences between successive images and velocity profiles in different years, we classified these glaciers into 12 surge-type glaciers and 16 non-surge glaciers including 4 advancing glaciers, 1 receding glacier, and 11 stable glaciers which did not show obvious changes in termini or surface features. Their characteristics, including areal changes, are listed in Table 1. E10, E15 and W13 are composite glaciers as recorded in the Second Glacier Inventory Dataset of China (Version 1.0). In general, our data show that glaciers on northeast-facing slopes are lower and more heavily debris covered than those on southwest-facing slopes (Table 1). The total glacier area has increased more than 1.33 km$^2$ since 1999.

Of the 17 glaciers that showed change over the study period, we measured the changes in terminus positions for 4 advancing glaciers, 1 receding glacier, and 4 surge-type glaciers, with the remaining 8 showing no clear terminus fluctuations (Fig. 3). As surge-type glaciers, the termini of W9, W12, and E10 moved more than 500 m over relatively short time periods. The durations of these surges ranged from a few months to several years. Even though W8 has not shown significant glacier terminus movements since 1999, it showed clear sign of surge behaviour in the images acquired by Landsat 5 between 1993 and 1999 (Fig. S1) and its terminus position change also shows a similar trend to the records of W12 and E10 after their rapid advances. For W13, we measured the change of the south branch glacier terminus. The main (west) branch terminus of W13 remained stable over the observation period; however, the east branch surged into the main branch at a rate of approximately 100 m/yr since 2007. The termini of advancing glaciers (W2, W3, W10, and W11) changed less than 300 m and did not change abruptly. They remained stable until 2007 and moved down-valley with a speed less than 30 m/yr thereafter. The only receding glacier (W5) in our study area diminished in size year-on-year between 1999 and 2016.

The comparison of interannual velocity fields allows us to understand the change of glacial motion patterns from 1999 to 2016 for non-surge glaciers and the motion change of surge-type glaciers before, during, and after surge events. Central velocity

profiles show the magnitude and timing of each glacier velocity field. It should be noted that our data focus on the glacier

ablation zones as the largely featureless accumulation areas contain few useful matches.

## 4.1 Non-surge Glaciers: Advancing Glaciers

Four advancing glaciers (W2, W3, W10, and W11) are located on the southwestern slopes of the KM with relatively less debris

cover than the glaciers on the northeastern slopes. Their surface features did not undergo significant change, except for a slight

displacement following the entire glacier with a total increase in area of 0.38 km$^2$ (Fig. 4).

The velocity profiles of these advancing glaciers show different motion patterns along their centrelines, but all with an obvious

acceleration during 2013–2016 compared with 1999–2002. For W2, W3, and W10 (Fig. 5a, b, and c), the values are ~10 m/yr

higher in the upper parts increasing to ~30 m/yr higher near the termini. For W11 (Fig. 5d), although the profiles in the upper

parts (km3 - km6) did not show a clear change, the profiles in the lower parts (km0 - km3) increased by ~30–50 m/yr.

## 4.2 Non-surge Glaciers: Receding Glacier

There is only one glacier (W5) that was detected to have receded over the study period, with its terminus position moving

~300 m up-valley, reducing its area by 0.31 km$^2$. W5 has two branches converging into the truck glacier from both north and

south slopes of the valley. A watercourse connected to the glacier terminus is clearly seen in the images acquired in July 2016

in an area previously occupied by glacier ice in September 2000 (Fig. 6). The images also show that the distribution of surface

debris cover also changed due to the variable flow of the three branches. The centreline velocity of W5 undulated between 20

m/yr and 40 m/yr prior to 2007, whereas from 2013 to 2016 the glacier was flowing consistently at 40 m/yr. The velocity of

the main glacier tongue showed an accelerating trend in the 2010s, despite the terminus position receding as described.

### 4.3 Non-surge glaciers: Stable Glaciers

Eleven glaciers in our study area are classified as being stable over the study period having shown no change in terminus

position or surface features. Most of these glaciers are located on the northeastern side of KM and are covered by heavy debris.

Velocity profiles of some glaciers (W1, E2, E3, E4) are absent in Fig. 7 because they exhibited no detectable motion and/or

5      decorrelation between available image pairs.

Stable glaciers in our study area show a range of glacial motion patterns. For E5, E6, and W7 (Fig. 7a, b, and g), the velocity

profiles in 2013-2016, compared to those in 1999-2003, show a decelerating trend (20 m/yr – 40 m/yr less). For E8, E14, and

E15 (Fig. 7c, d, and e), the velocities in 2013-2016 remain unchanged when compared to 1999-2003. W4 is the slowest glacier

among the stable glaciers with peak velocity less than 30 m/yr (Fig. 7f), although its velocity at km0 - km3 increased ~20 m/yr

10     in 2013-2016. Even though the glacier termini and surface features remained stable, their motion change is diverse.

### 4.4 Surge-type Glaciers

We identify 12 surge-type glaciers and 14 discrete surge events from the studied glaciers. These surge events generally have

two styles of behaviour based on whether the surge impacts the terminus or not.

5 glaciers have surged resulting in a terminus advance. From 1999 to 2016, these glacier termini advanced only once.

15     According to glacier termini evolution in Fig. 3, E10, W9, and W12 advanced more than 500 m with the surge duration varying

from a couple of months (W9) to several years (W12). W12 is unique in that it does not have an apparent terminus that can be

interpreted from satellite images. We take its latest change in debris downstream as its terminus position. The south branch of

W13 advanced ~450 m to the north and compressed the trunk glacier by ~150 m (Fig. 8). Subsequently, the trunk branch of

W13 advanced ~100 m every year since 2008 (Fig. 3). The surge of W8 was in its recession stage in 1999 with the beginning

20     of terminus advance in 1993.



There are 7 glaciers that surged without affecting their termini. These glaciers showed obvious and dramatic changes in surface features within relatively short time periods (several months) from 1999 to 2016. Figure 8 shows the comparison before and after the surge events of E7, E11, E12, and E13, E1, E9, and W6 are given in Fig. S1. These glaciers are heavily debris-covered and most develop what appear to be composite masses of rock and ice at their termini. In most cases the surges destroy the

original surface structures and reshape the glaciers and debris during the surges. The satellite images of E11, E12, and E13 clearly show that surge fronts propagated down-glacier and transported mixtures of ice and debris creating new alluvial fans, which in addition changed the original glacial outflow. But for E1 and E7, the surges did not destroy the original surface features and the general appearance remained after the surge events. Although these surges did not result in terminus advances, they modified the inner surface structures of glaciers, bringing about the growth of new debris-dammed supraglacial ponds.

Centreline profiles of annual velocities provide quantitative information about the glacial motion before, during, and after the surges. For E1, E7, and E12 we could only extract the profiles during the active phases showing a clear surge front in each case (Fig. 9). Due to the limitation of available satellite images, it is not possible to provide complete velocity profiles throughout the other glacier surge events. Instead, we present the comparison before and after the surges to identify the impact of surges on glacial motion. Profiles of E1 reveal two surge events: the first peaked between 1999 and 2000, and the second

peaked between 2013 and 2014. The interval of the surge events is ~14 years which can be taken as an indication of the surge cycle period of E1. Two different surge fronts were also detected on the surface of E11 with a 14-year surge cycle from 1999 to 2013 (Fig. 8). Since the width of E11 (most parts) is less than 300 m it is difficult to detect the displacement fields directly from satellite images with low spatial resolution. The two surges of E1 are not identical in their nature. According to Fig. 9a and b, the surge in 1999 affected a larger area (km2 – km7) with higher peak velocity (~100 m/yr) than the surge in 2013

(affected area: km1 – km5, peak velocity: ~70 m/yr). E7 (Fig. 9c) has a much higher peak velocity, nearly ten times greater during surge active phases (~150 m/yr) than in its quiescent phase (~15 m/yr). The profiles of E12 show a moving surge front propagating downstream, although our observations are limited to the initiation and decay of the surge (Fig. 9f). In the cases of E1, E7, and E11, the surge lasted for between 3 and 4 years. For E12, the surge front is still detectable near its terminus,

lasting 8 years after initiation. Judging from the terminus changes shown in the imagery, the surges of W8 and W12 appeared to last 6-7 years and surges of W9 and E10 lasted 1-2 years.

Changes in the surface features of E9 and W9 are also indicative of surge behaviour in 2007. The surface velocity of E9 (2013 to 2016) appeared to be the same as prior to the surge in 1999-2003, except in the lower reaches, where the glacier showed

faster flow in the earlier image pairs (Fig. 9d). The velocity profiles of W9 (Fig. 9j) in 2013–2016 show a similar trend to those in 1999–2002, except for a higher peak velocity zone at 3–4 km from terminus (~90 m/yr) shown in 2013–2016. We suggest that until 2013, E9 and W9 had recovered from the surges in 2007 judging from the similar motion patterns in 1999-2003 and 2013-2016. Profiles of E10, E13, W8, and W12 vary markedly in both amplitude and shape. Most parts of these glaciers flowed much slower between 2013 and 2016 than they did between 1999 and 2003. W12 increased its velocities year-on-year in 2013-

2016 indicating it is currently recovering from the last surge event (Fig. 9k). E10, E13, and W8 may follow this pattern in the near future as well. For W13 (Fig. 9l), the south branch surged into the trunk glacier at km2 from the terminus in 2008. The trunk glacier was compressed by the south branch after the surge, which resulted in a motion change of the trunk glacier, characterized by acceleration in the lower parts (km0 – km2).

Velocities of W6, which surged before 1989 (Fig. S1), were 20–60 m/yr higher in 2013–2016 than those in 1999–2002 and

the profiles show a clear downstream-propagating trend (Fig. 9h). The retreat in the ablation zone recent years and the fast-moving mass in the accumulation zone indicate that W6 is preparing for the next surge and its surge cycle period is more than 27 years (1989-2016).

## 5 Discussion

### 5.1 Motion Pattern of Surge-type Glaciers

The existence of at least twelve surge-type glaciers out of 28 in the KM provides strong evidence that the Pamir region is one of the world's most active surge zones, as indicated by the geodatabase of surge-type glaciers (Sevestre and Benn, 2015). By

analysing the velocity profiles of these glaciers in the study area, we define 12 glaciers (E1, E7, E9, E10, E11, E12, E13, W6, W8, W9, W12, and W13) as surge-type glaciers. 14 surge events were identified with E1 and E11 surging twice each until 2016. Despite these glaciers being of different sizes and shapes, we find that they tend to show broadly similar motion patterns during their active and quiescent phases (Fig. 10). Prior to the onset of the surge, the glacier occupies a relatively stable stage (Profile-a). In the active phase (Profile-b to Profile-d), the glaciers in our study area show peak velocities in the first year (E1 and E7) when surges occur, followed by gradually decreasing velocities in the following few years as the surge front propagates downstream. After the active phase, the glacier surface velocities drop to their minimum level (Profile-e to Profile-g). In many cases, the glacier tongue will then remain largely stagnant flowing 0–10 m/yr in most parts for several years as ice is recharged in the accumulation zone (cf. Fig. 9i). After an uncertain period (varying from 11 years to 14 years) of accumulation, the glacier again begins to transport ice downstream in a gradually recovered velocity initiated from the accumulation zone to ablation zone until it reaches its stable stage before the next surge (cf. Figs. 9d and j). We interpret the relatively motionless (Profile-e to Profile-g) and stable (Profile-a) stages to define the glacial quiescent phase. We note that the exact periods corresponding to the active and quiescent phases differ with glacier geometry and amount of debris-cover.

Previous studies have reported traveling waves during glacier surges, such as observations of Gasherbrum Glacier (Mayer et al., 2011) and Kunyang Glacier (Quincey et al., 2011). They are interpreted to be a result of both hydrological (Kamb et al., 1985) and thermal changes (Fowler et al., 2001). Hydrologically controlled surges are suggested to accelerate during winter condition when the downglacier basal hydrology is inefficient and decelerate during summer conditions when the water pressure is reduced by melting-water channelization (Raymond, 1987). The hydrologically controlled surge front represents the boundary between an efficient tunnel drainage system in upglacier and an inefficient cavitized system in downglacier. Thermally controlled surges rely on a change in conditions at the bed and the surge fronts in this case represent the transition between warm ice (upglacier) and cold ice (downglacier) (Clarke, 1976). Another explanation to the fronts of surges in the Karakoram can relate to the individual glacier configurations (cf. Quincey et al., 2015). Slow-moving ice in downglacier,

which could be immobile cold ice or a remnant of a previous glacier surge depending on different glaciers, exists as an obstacle to fast-moving ice in upglacier.

Although many other surge events remain undiscovered, we identified seven glaciers (that surged nine times in the KM (E1, E7, E9, E11, E12, E13, and W6)) that occurred without affecting their termini. Due to their heavy debris cover and long glacier

tongues these events were inhibited by immobile and likely cold ice, and consequently these glacier termini remained stable during and after the surge events. The surges of E1 and E7 showed minor surface feature advances and the other five surges (E9, E11, E12, E13, and W6) caused the rearrangement of surface features in the middle parts of the glacier tongue, as well as overriding the remnant left by a possible previous glacier surge, similar to the surges of Braldu and Unnamed1 Glaciers in the Karakoram (Quincey et al., 2015). This type of glacier surge is common in eastern Pamir (Lv et al., 2016; Shangguan et al.,

2016) although their evolutions differ both spatially and temporally. We suggest these flow instabilities can be contributed to the geomorphological characteristics of different glaciers, such as the slope of the entire glacier, the size of accumulation area, the width of the outlet, and so on.

Even though the poor temporal resolution of our data cannot provide any seasonal signal of the surge evolution, it is clear that most of the surges reached their peak velocities in the first year of initiation and their termination phase lasted for years,

contrasting with the hydrologically controlled surges in Alaska where the termination phase has been shown as much more abrupt than the initiation phase, lasting as little as several days (Burgess et al., 2013).

We attribute the relatively shorter surge cycle period of E1 and E11 compared to other glaciers to their large accumulation areas and abundant ice supply. E1 has one trunk glacier and three tributaries and E11 has a narrow outlet, which make them much quicker to accumulate and transport new-formed ice downstream. On the other hand, W6, which appears to have a cycle

period of more than 27 years, has a winding and broad outlet and only one reservoir in its accumulation area. The fast-moving mass (70 m/yr) in recent years indicates that W6 is approaching its next surge. Our data suggest that return periods vary widely from glacier to glacier but in general the larger the accumulation area and the narrower the outlet, the shorter return period the glacier has. The return period of glacier surges in Kingata appear to range between one decade and several decades, which is

coincident with Karakoram glacier surges are reported to be of the order of several decades (Quincey and Luckman, 2014; Quincey et al., 2015). This is quite different from other thermally controlled surges elsewhere with a return cycle up to several centuries (Dowdeswell et al. 1991).

The duration of Kingata glacier surges is usually less than ~10 years which is taken as the duration of surges in Svalbard, ranging from 1 year to 8 years. Indeed, most Kingata surges last 3-4 years, similar to those in the Karakoram. The extreme short-lived surges, and the large-scale velocity variations of W9 and E10 (1-2 years), are similar to surge events reported on Shakesiga Glacier in the Karakoram (Quincey et al., 2015) and Kelayayilake Glacier on Mont Tobe Feng (Lv et al., 2016). Kelayayilake Glacier, located near the south of KM, initiated in winter months and terminated in summer months. The significantly-reduced water discharge before and during the surge reported by Gez hydrological station implicate this type of glacier surge is partially dominated by a hydrological control. Unfortunately, there are no hydrological data for the KM to help resolve the dominant processes operating here in a similar way.

Glacier surges in eastern Pamir share many, although not all, common features with Karakoram glacier surges and they both do not completely resemble either thermal or hydrologically controlled surges reported in Svalbard and Alaska. We suggest that surges in eastern Pamir, as with surges in Karakoram, have variable controlling processes depending on the thermal and hydrological conditions as well as the geomorphological characteristics of different individuals.

Although such surge-type glaciers represent a small percentage of the world's glacier population, they are of great importance in the investigation of glacier processes, flow instabilities, and fast glacier flow (Clarke, 1987). Before and after the surges, the interannual velocity profiles of surge-type glaciers show significant variations in their shape and amplitude, which reflect the internal structure change. The vigorous motion processes of such a large number of surge-type glaciers give us a different insight into the acknowledged stable glacial situation in the Pamir region in the context of glacial shrinkage in HMA; in addition, more surge events remain undiscovered because of insufficient available high spatial resolution satellite data and decorrelation between image pairs that precludes the derivation of velocity data by feature tracking.

## 5.2 Glacial Motion Pattern Change

Glaciers in northwestern China are of continental type, formed under dry and cold climate conditions. These glaciers were previously regarded as stable, low-ablation glaciers with short glacier tongues and low motion rates. In particular, glaciers in the Pamir region experienced the least glacial retreat in HMA over recent decades (Yao et al., 2012). In our study, with the exception of surge-type glaciers, glaciers both advanced and receded, showing diverse motion pattern changes during the study period.

The patterns we describe in this study show some similarity with glacial motion patterns in the Bhutan Himalaya; Kääb (2005) ascertained that large differences in dynamics were present between fast-moving north-facing glaciers and slow-moving south-facing glaciers and concluded that different erosion and sediment evacuation processes should act on the two sides of Bhutan Himalayan main ridge. There, glaciers on the northern slope are debris-free and glaciers on the south slope have heavy debris cover with thermokarst features. In the KM, glaciers on the northeastern slopes flowing into the Tarim Basin range in peak velocity from 30 m/yr to 60 m/yr whereas fast-moving glaciers on the southwestern slopes flowing into the Pamir Plateau range from 40 m/yr to 120 m/yr. The topography and surface features are very different either side of the divide, which we suggest might lead to the motion variation we have detected here.

Elevation profiles (Fig. 11a) indicate that glaciers to the southwest have plateau-type accumulation areas and gentle surface slopes, which might exert a positive influence on the formation and preservation of new ice in the upglacier zones. The westerly winds deliver most of the total annual precipitation to eastern Pamir (Su et al., 1989; Seong et al., 2009; Jiang et al., 2014). As the wind-facing slopes, glaciers on the southwest have the potential to capture most of the precipitation. Little precipitation makes its way over the divide to nourish the high-elevation areas on the northeast, even though snow may be blown further into the accumulation area by winds from the southwest. Most glaciers on the northeastern slopes have steep-ice accumulation zones, indicating that they are predominantly fed by avalanches. No reliable measurements of precipitation exist in this region with which we can verify these trends. However, the southwestern glaciers have bigger total glacierised areas than the northeastern ones judging from the mean hypsometry profiles of glaciers on both slopes (Fig. 11 b and c). Meanwhile, the



southwestern glaciers have most ice at high elevation, which is opposite for the northeastern ones. The abundant accumulation zones on the southwest give a reasonable explanation for their faster moving behaviour compared to the other slopes. Glaciers on the northeastern-facing slopes also flow down to much lower elevations, usually about 1000 m lower than those on southwest (Fig 11a). Some of these glaciers flow at a much slower rate and tend to remain stable with well-developed

supraglacial melt ponds (cf. E3), features which are also prevalent for many other long debris-covered glacier tongues in HMA, including the Everest region (Luckman et al., 2007; Scherler et al., 2008; Quincey et al., 2009; Watson et al., 2016). These glaciers in different catchments around the Everest region are experiencing substantial mass loss over the last few decades as revealed by DEM differencing (King et al., 2017). Although we have no ice mass balance data to support our interpretation, based on visual evidence we would suggest that KM glaciers on the northeastern slopes are also in a period of rapid recession.

By comparing the interannual glacier velocity profiles, we can determine how the glacial motion patterns changed. Fast moving glaciers on southwestern slope are most likely to dynamically adjust to climate variations. The high velocities reflect an efficient ice mass turnover within glaciers, and acceleration is coincident with the positive mass balance reported in the Pamir region (Gardner et al., 2013; Gardelle et al., 2013; Osmonov et al., 2013; Holzer et al., 2015; Kääb et al., 2015; Brun et al., 2017). For the receding glacier (W5 in Fig. 7), the glacial motion in the 2010s shows an increasing or, at least, stable trend.

We suggest that its retreat is the consequence of the interaction between the trunk glacier and its two branches.

Changes in glacial motion can also be attributed to changes in precipitation and temperature over recent decades (Raper and Braithwaite, 2009). Most glaciers in the KM are relatively small compared to glaciers in nearby areas, such as Kalayayilake Glacier (Lv et al., 2016; Shangguan et al., 2016) and Muztag Glacier (Yang et al., 2013). No in-situ climate data exist for this region making it difficult to disentangle glacier response to a change in accumulation from other factors. Moreover, there are

no robust data on ablation rates or ice thickness that might be used to make estimations of glacier response times (Jóhannesson et al., 1989). Nevertheless, it may be reasonable to suggest that the acceleration/deceleration we observed is a product of climate changes over the last two decades given their relatively small size and insulating debris cover. This speculation

underlines the need for more detailed meteorological records as well as robust and long-term field data if the climate impacts on glaciers in the KM and surrounding regions are to be robustly characterised.

## 6 Conclusions

Using cross-correlation feature tracking on Landsat and ASTER images, we have studied the temporal and spatial evolution of glacial motion in the KM for the first time. When combined with the record of terminus positions and surface features indicative of surging, these data demonstrate that: (1) By analysing and quantifying glacial motion in the KM from 1999 to 2016, we can define 12 surge-type glaciers and summarize a surge-type glacial motion pattern for the characterization of different surge states. (2) The controlling process of each surge varied on different individuals depending on their thermal and hydrological conditions and geomorphological characteristics. These surges resemble those in Karakoram and do not fit into classic thermal and hydrological classification. (3) Glaciers on southwestern slope have relatively higher speeds than glaciers on northeastern slope. (4) Glaciers in KM experienced a significant and diverse change in their motion patterns which may indicate decadal scale changes in climate, although glaciers in the Pamir region are acknowledged of stable state in mass balance and show the least glacial retreat in HMA. In future research, precipitation and temperature data, associated with estimates of glacier thickness and its change, will greatly aid the study of climate impacts on glaciers in the Pamir region.

## 7 Data availability

The Landsat-7, Landsat-8, and ASTER images can be freely downloaded from http://glovis.usgs.gov. ASTER GDEM can be found from NASA's Earth Observing System Data and Information System (EOSDIS). The Second Glacier Inventory Dataset of China is available from http://westdc.westgis.ac.cn/data/f92a4346-a33f-497d-9470-2b357ccb4246. These data are publicly accessible.

**Author contributions**

LV Mingyang, LIU Guang, GUO Huadong and LU Xiancai designed this study. LV Mingyang carried out the data processing. LV Mingyang, Duncan J. Quincey, and YAN Shiyong wrote the paper. RUAN Zhixing and DING Yixing assisted in data interpretation and edited the paper.

5 **Competing interests**

The authors declare that they have no conflict of interest.

**Acknowledgments**

This research was supported by the Strategic Priority Research Program of the Chinese Academy of Sciences [XDA19070202]; the International Cooperation Program of CAS [131211KYSB20150035]; National Natural Science Foundation of China 10 (NSFC) [41120114001, 41001264]; Natural Science Foundation of Jiangsu Province [No.BK20150189]; The Landsat-7, Landsat-8, and ASTER images were freely downloaded from http://glovis.usgs.gov. ASTER GDEM can be found from NASA's Earth Observing System Data and Information System (EOSDIS). We acknowledge support from Owen King on the analysis of glacier hypsometry.

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





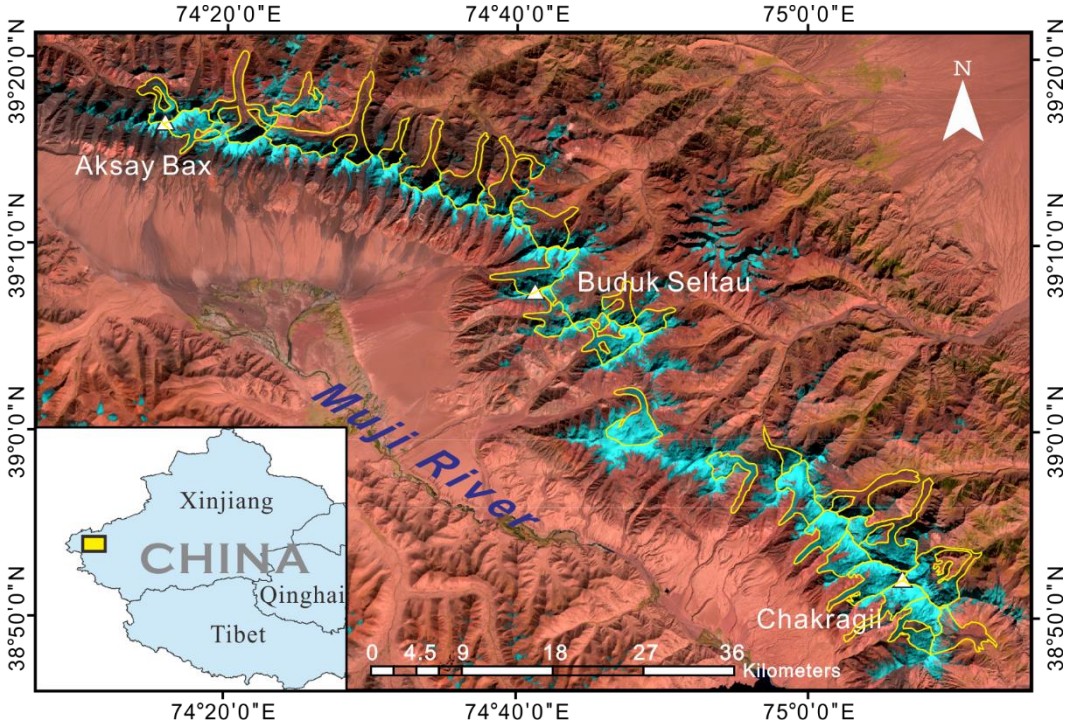

**Figure 1: Landsat OLI image (bands 6-5-4) of the Kingata Mountains acquired on October 3rd, 2014. Light blue areas represent bare ice, red-brown areas represent bare soil and rocks, and green areas represent grass or other vegetation. Glaciers with area greater than 5 km² are highlighted by yellow lines.**



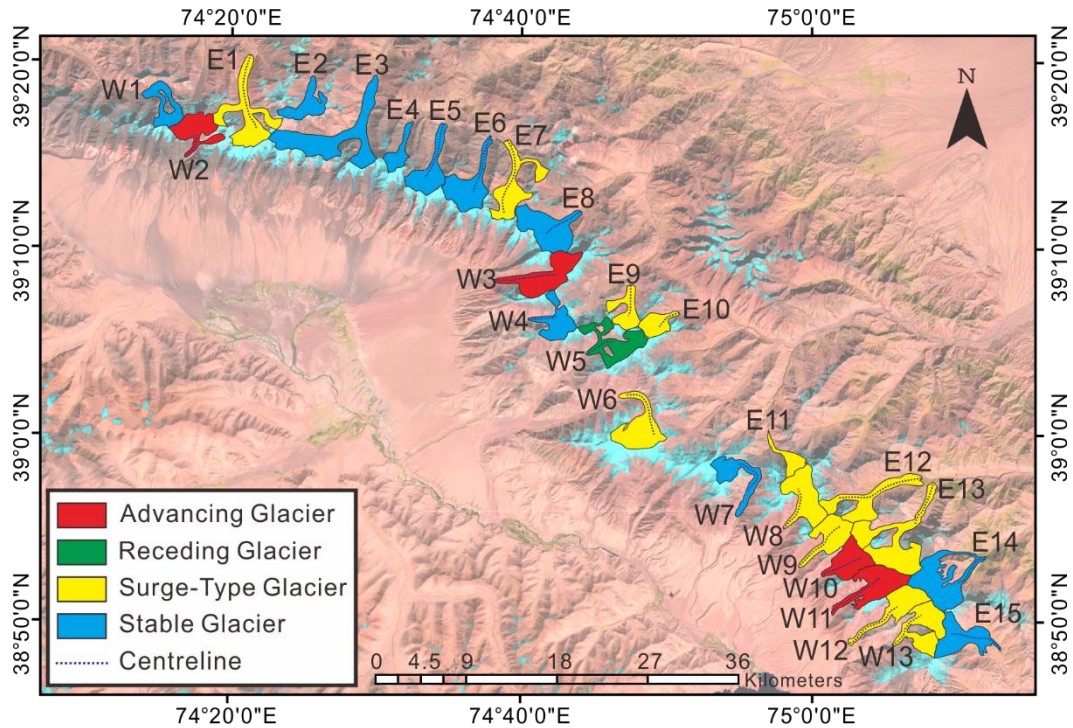

**Figure 2: Studied glaciers and their numbers in this study.**

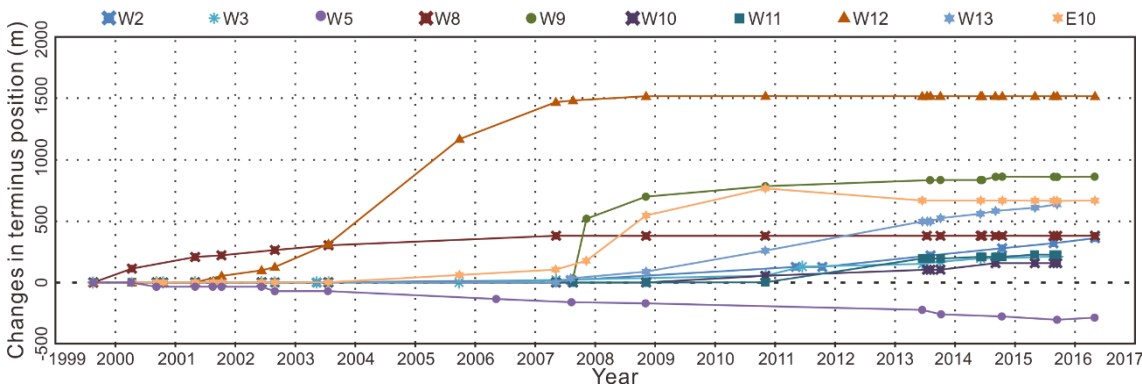

5    **Figure 3: Changes in glacier terminus positions from 1999 to 2016 for 10 glaciers with detectable change.**



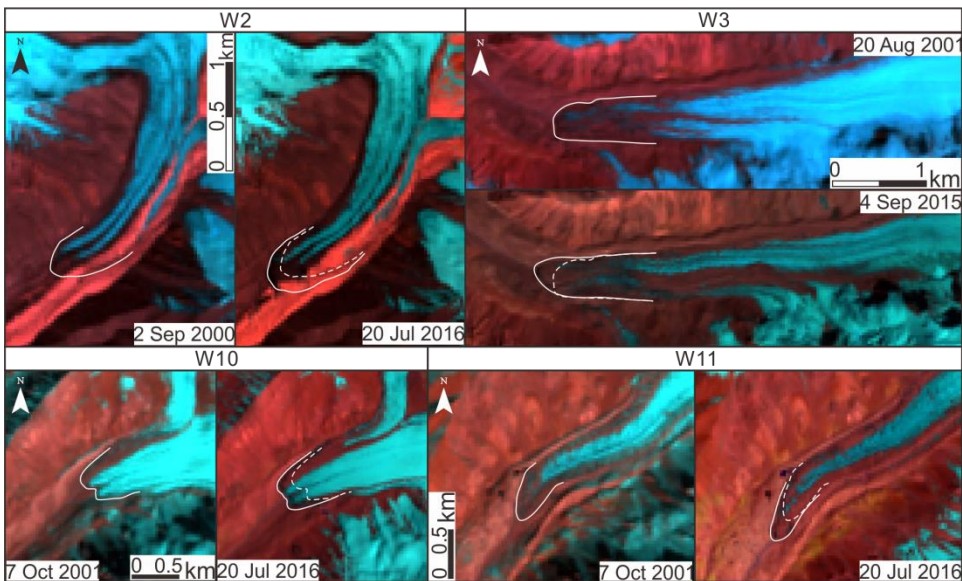

**Figure 4: Comparison of the advancing glaciers before and after the advance. The white lines show the positions of glacier termini in the images and the dashed lines show the relative positions of the termini in the associated pre-advance images.**

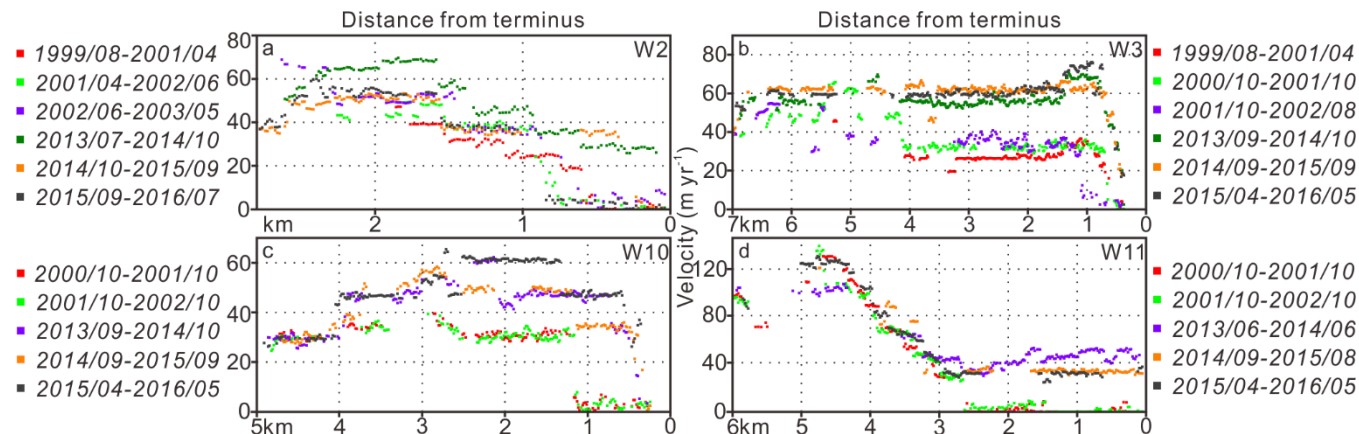

**Figure 5: Centreline annual-velocity profiles for advancing glaciers: (a) W2, (b) W3, (c) W10, and (d) W11.**



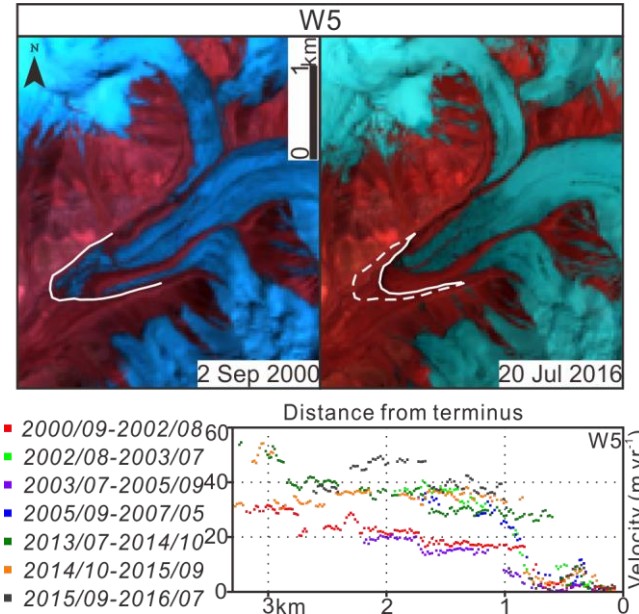

**Figure 6: Comparison of the receding glacier W5 before and after the retreat and its centreline annual-velocity profiles.**





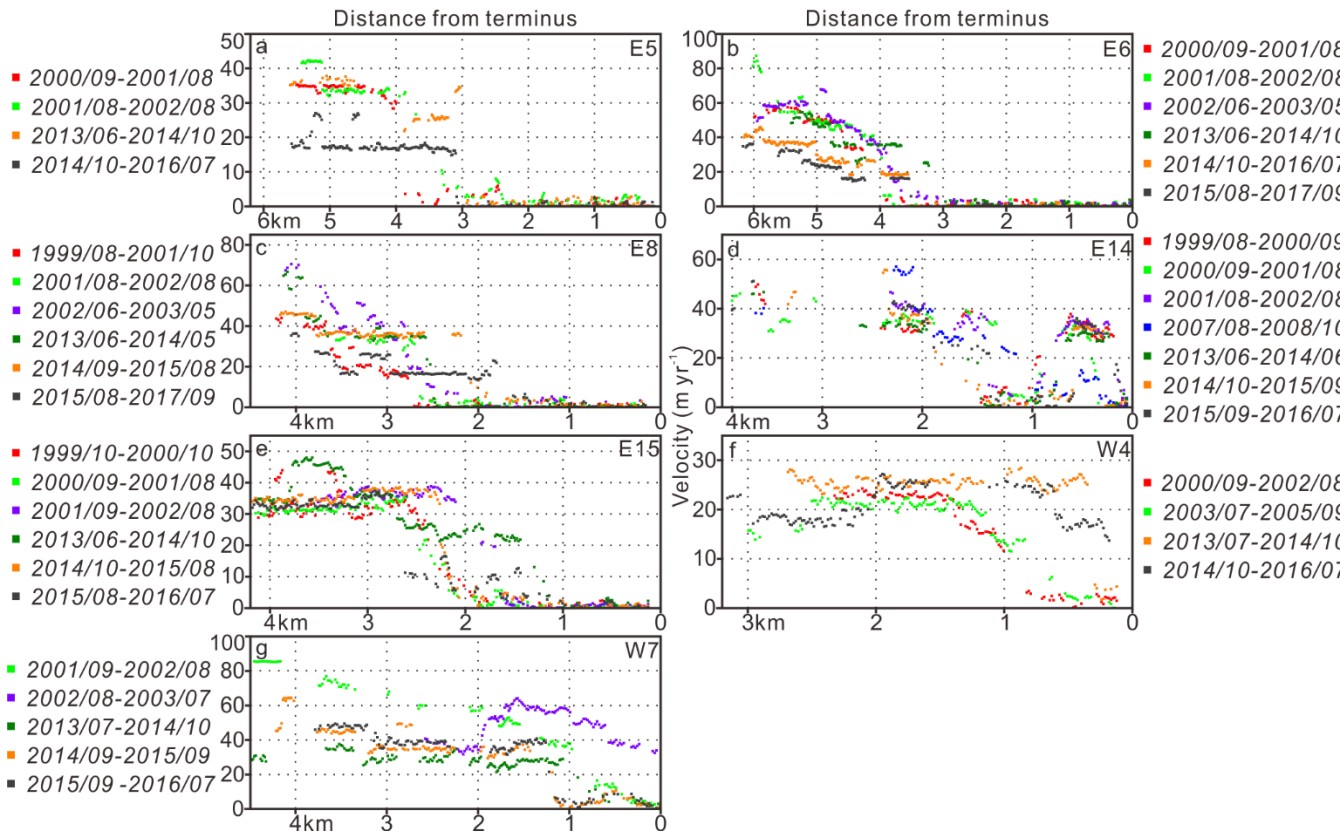

**Figure 7: Centreline annual-velocity profiles for stable glaciers: (a) E5, (b) E6, (c) E8, (d) E14, (e) E15, (f) W4, and (g) W7. Note that profiles of W7 start from the clean ice boundary rather than the glacier ternimus.**





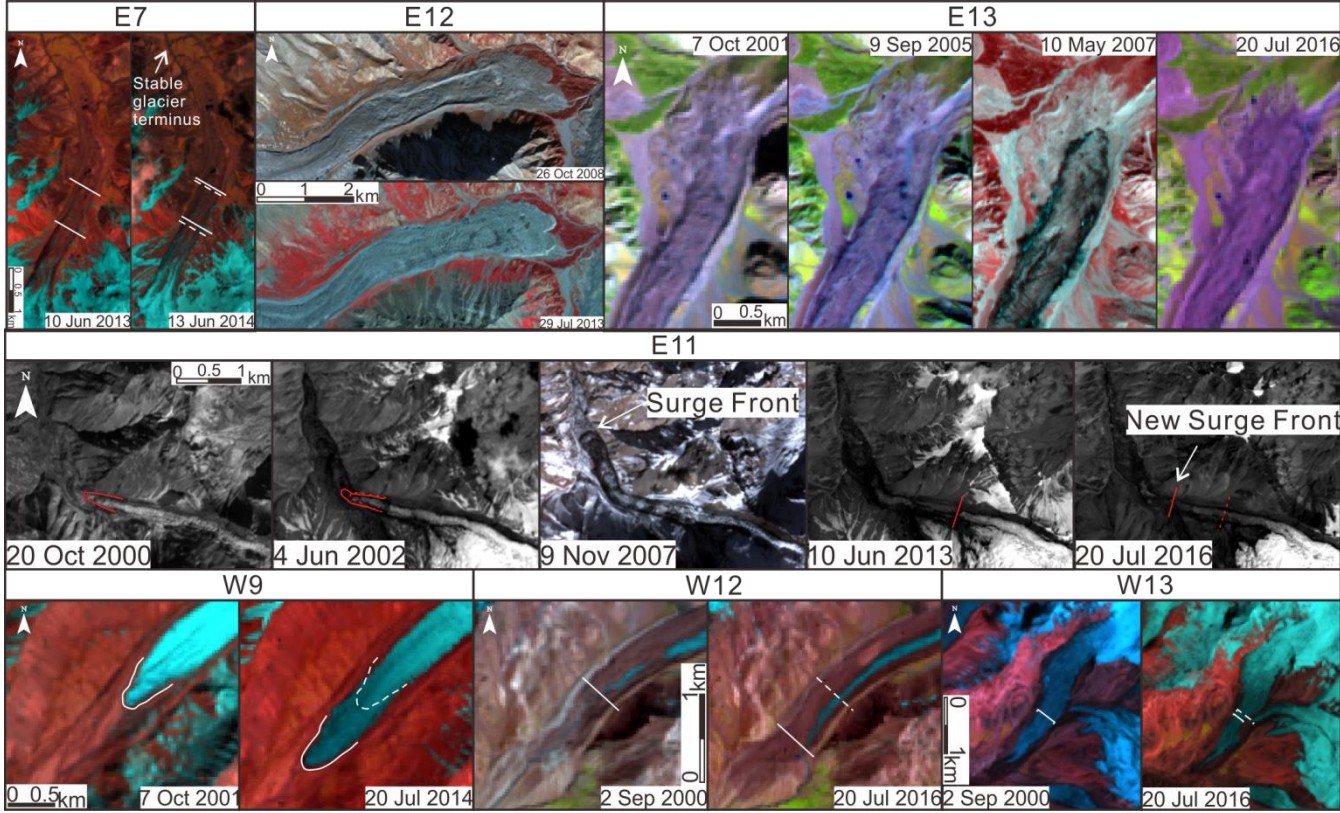

**Figure 8: Comparison of the surge-type glaciers before and after surge events. Note that the south branch of W13 surged into the trunk branch and narrowed the trunk branch to nearly two-thirds of its original width. E11 is given the panchromatic band of Landsat, except ASTER true colour combination in 2007.**





**Figure 9: Centreline annual-velocity profiles for surge-type glaciers: (a) E1 (1999-2011), (b) E1 (2007-2016), (c) E7, (d) E9, (e) E10,**

**(f) E12, (g) E13, (h) W6, (i) W8, (j) W9, (k) W12, and (l) W13.**





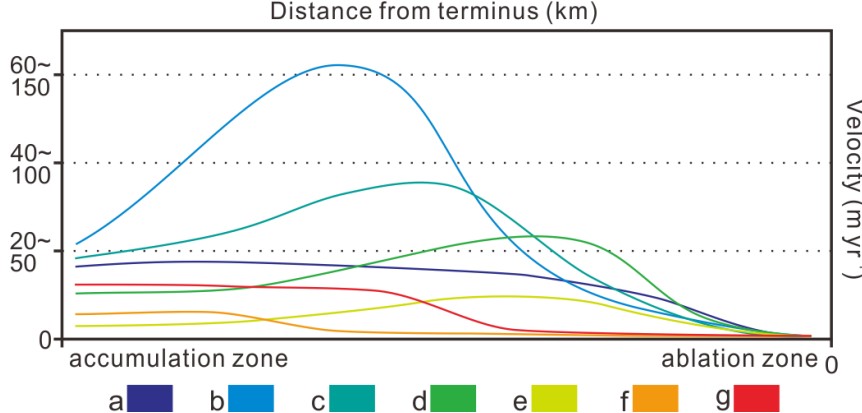

**Figure 10: Conceptual characterisation of the glacial motion pattern of surge-type glaciers in the Kingata Mountains**





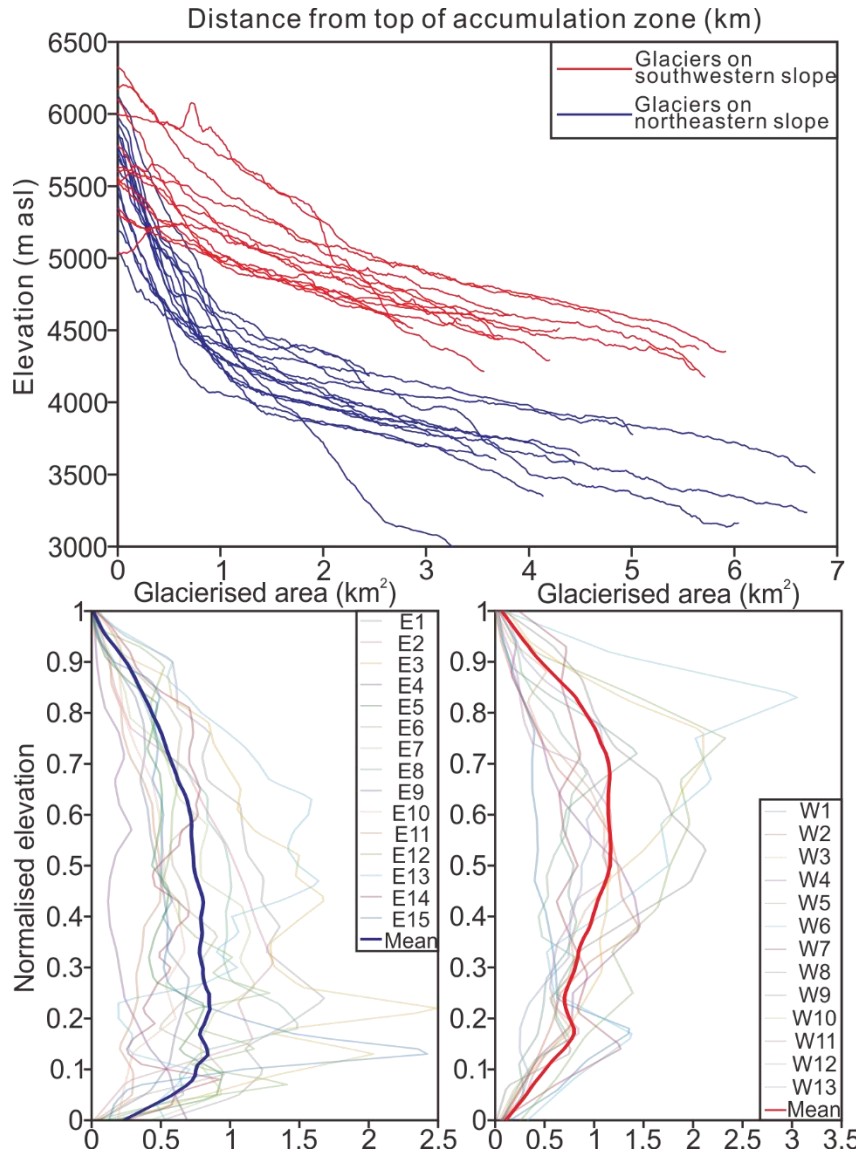

**Figure 11: (a) Surface elevation profiles of studied glaciers extracted from ASTER GDEM, (b) hypsometry curves of northeastern glaciers and (c) hypsometry curves of southwestern glaciers.**

5      **Table 1: Attributes of the glaciers in this study**





| No. | Area (km2) | Debris cover (%) | GLIMS_ID | Lon. | Lat. | Max/Min elevation (m) | Mean elevation (m) | Mean slope (°) | Mean aspect (°) | Area changed (km2) |
|---|---|---|---|---|---|---|---|---|---|---|
| E1 | 24.11 | 26.3 | G074348E39282N | 74.35 | 39.28 | 5912.2/3796.4 | 4760.2 | 25.4 | 22.5 | |
| E2 | 7.37 | 19.7 | G074419E39295N | 74.42 | 39.29 | 5423.4/4223.1 | 4707.3 | 20.5 | 16.8 | |
| E3 | 25.52 | 46.0 | G074449E39260N | 74.45 | 39.26 | 5836/3829.2 | 4606.2 | 26.5 | 42.5 | |
| E4 | 5.25 | 38.5 | G074525E39252N | 74.52 | 39.25 | 5913.6/3943.3 | 4570.4 | 30.6 | 40.2 | |
| E5 | 9.29 | 42.7 | G074560E39231N | 74.56 | 39.23 | 5692.7/3716.9 | 4563.7 | 30.3 | 25.7 | |
| E6 | 13.65 | 29.3 | G074606E39223N | 74.61 | 39.22 | 5774.1/3584.2 | 4562.8 | 26.5 | 42 | |
| E7 | 10.75 | 52.1 | G074652E39213N | 74.65 | 39.21 | 5828.2/3744.1 | 4525 | 24.2 | 6.4 | |
| E8 | 17.17 | 21.4 | G074696E39186N | 74.70 | 39.19 | 5895.3/3675.5 | 4655.2 | 29.9 | 59.1 | |
| E9 | 6.43 | 42.8 | G074785E39097N | 74.79 | 39.09 | 5704.5/4016.4 | 4780.6 | 27.6 | 29.4 | |
| E10 | 5.53 | 3.3 | G074826E39097N | 74.83 | 39.10 | 5629.4/4129.4 | 4884.3 | 28.8 | 49.2 | 0.22 |
| | | | G074813E39103N | 74.81 | 39.10 | 5631.6/4497.2 | 4948 | 29 | 54.3 | |
| E11 | 5.26 | 8.6 | G074981E38974N | 74.98 | 38.97 | 5788.9/4010.1 | 4983.2 | 26.4 | 2.3 | |
| E12 | 14.46 | 60.2 | G075050E38938N | 75.05 | 38.94 | 5900.8/3336.1 | 4534.2 | 29.3 | 54.3 | |
| E13 | 23.00 | 11.9 | G075083E38897N | 75.08 | 38.90 | 6095.5/3139.3 | 4780.4 | 28.3 | 51 | |
| E14 | 21.84 | 11.8 | G075145E38873N | 75.15 | 38.87 | 6592.4/2843.1 | 4547.7 | 34.2 | 71.4 | |
| E15 | 12.78 | 46.9 | G075160E38813N | 75.16 | 38.81 | 6106.5/3730.5 | 4714.4 | 32.6 | 107.6 | |
| | | | G075150E38828N | 75.15 | 38.83 | 6394.3/3894.9 | 5239 | 42.2 | 124.3 | |
| W1 | 9.62 | 23.2 | G074258E39293N | 74.26 | 39.29 | 6059.7/4421.7 | 5030.7 | 25 | 342.2 | |
| W2 | 12.27 | 3.1 | G074295E39269N | 74.29 | 39.27 | 6037.1/4529.6 | 5254.5 | 24.6 | 220.8 | 0.14 |
| W3 | 15.05 | 6.0 | G074696E39143N | 74.70 | 39.14 | 6069.7/4383 | 5332.8 | 25.1 | 313.2 | 0.08 |
| W4 | 9.18 | 5.3 | G074707E39103N | 74.71 | 39.10 | 6015.6/4544.7 | 5229.6 | 24.1 | 308.1 | |
| W5 | 13.20 | 2.6 | G074774E39084N | 74.77 | 39.08 | 5811.5/4616.1 | 5257.8 | 21.1 | 299.7 | -0.31 |



| | | | | | | | | | | |
|---|---|---|---|---|---|---|---|---|---|---|
| W6 | 14.27 | 7.0 | G074800E39007N | 74.80 | 39.01 | 5634.7/4487.9 | 5149.1 | 12 | 24.8 | |
| W7 | 10.19 | 17.1 | G074911E38969N | 74.91 | 38.97 | 5625/4231.7 | 4926.2 | 16.7 | 103.2 | |
| W8 | 12.69 | 1.4 | G074990E38942N | 74.99 | 38.94 | 5906/4456 | 5236.6 | 23.9 | 260.4 | 0.05 |
| W9 | 10.61 | 2.2 | G075022E38910N | 75.02 | 38.91 | 6102/4421 | 5394.4 | 21.3 | 247.3 | 0.04 |
| W10 | 9.84 | 1.3 | G075043E38887N | 75.04 | 38.89 | 6073.5/4581.5 | 5302.3 | 23 | 271.2 | 0.09 |
| W11 | 13.38 | 3.8 | G075075E38868N | 75.07 | 38.87 | 6576.7/4454.9 | 5460.1 | 25.7 | 288.6 | 0.07 |
| W12 | 10.51 | 10.8 | G075097E38848N | 75.10 | 38.85 | 6613.2/4118 | 5365.3 | 25.7 | 237 | 0.50 |
| W13 | 13.97 | 2.4 | G075126E38839N | 75.13 | 38.84 | 6438.5/4270.8 | 5558.5 | 24.4 | 254.6 | |
| | | | G075128E38819N | 75.13 | 38.82 | 6155.3/4581.6 | 5328.1 | 29.6 | 296.8 | |