# Peer review of "Characterising the behaviour of surge and non-surge type glaciers in the Kingata Mountains, Eastern Pamir, from 1999 to 2016"

_The Cryosphere, 2018_

## Referee Comment (RC1) · Anonymous Referee #1 · 21 Sep 2018

**Review of tc-2018-131, Sept. 21, 2018**
Characterising the behaviour of surge and non-surge type glaciers in the Kingata Mountains, Eastern Pamir, from 1999 to 2016
Mingyang Lv, Huadong Guo, Xiancai Lu, Guang Liu, Shiyong Yan, Zhixing Ruan, Yixing Ding, and Duncan J. Quincey

P2, L10: it would also be useful to mention the nearby Karakoram here (which is already described in several of the papers you reference). An added reference to this paper would also be useful:
Gardelle et al. 2012. Slight mass gain of Karakoram glaciers in the early twenty-first century. *Nature Geoscience*, 5, 322-325

P2, L13: clarify the period that the reference to positive mass balance of the eastern Pamir refers to

P4, L14: the distance from Taxkogan meteorological station to the Kingata Mountains is >100 km, so this should be noted

P4, L16: clarify where a 'mean temperature as high as $15^{\circ}C$' is referring to: e.g., over glaciers? At which altitude?

P5, L4: I think that 'high accuracy' is optimistic given the resolution of your imagery; 'good accuracy' would seem to be a better descriptor

P6, L4: state what the 'common reference dataset' was

P6, L10: state which band(s) and resolution were used for the determination of glacial motion; in part, this is needed to understand the pixel size being referred to here

P7, L6: state the averaging distance used for the velocity smoothing, and how blunders were identified in the data

P7, L7: how do you know that the accuracy of manual measurements is within 30 m, and therefore which area changes can be considered significant? There are some useful papers that address this issue directly, such as:
- Hall, D. K., Baa, K. J., Schöner, W., Bindschadler, R. A., and Chien, J. Y. L., 2003: Consideration of the errors inherent in mapping historical glacier positions in Austria from the ground and space (1893–2001). Remote Sensing of Environment, 86: 566–577.
- Paul, F., and 19 others, 2013: On the accuracy of glacier outlines derived from remote-sensing data. Annals of Glaciology, 54(63): 171–182.

P7, L18: be more explicit as to what 'discrete changes in glacial motion' refers to

P8, L9: state what the total glacier area was in 1999 so that the 1.33 $km^2$ change can be put in perspective

P8, L13: can you be more specific about the duration of the individual surges? A range of 'a few months to several years' is pretty broad

P8, L14: please better describe what the 'clear sign of surge behaviour' was – e.g., how far did the glacier advance over which period? Fig. S1 is too small and too low resolution to make out much meaningful detail (also see specific comments below). Please improve!

P8, L18: please specify the end period for the movement of '100 m/yr since 2007'. Also specify the period that 'termini of advancing glaciers… changed less than 300 m' refers to: my initial assumption was 1999-2016, but in the next sentence you say that remained stable until 2007

P9, L6: I don't understand what 'displacement following the entire glacier' means. It would also be useful to state what the change in glacier length was, in addition to area

P9, L13: change 'truck glacier' to 'trunk glacier'!

P9, L15: to me, W5 looks like a surge-type glacier due to the distorted moraines in lower terminus region in 2000 (in particular it looks as if the tributary to the SE might have recently surged; see Google Earth for high resolution imagery). It would be worthwhile looking through old Landsat imagery to see if this surge was captured, which would prove this definitively. I would therefore argue that the unusual recession of this glacier is primarily due to the quiescent phase of the surge cycle, which would make sense when no other glacier in the KM retreated over the study period.

P10, L2: you mean that these glaciers showed no detectable change beyond error limits? This is why the errors need to be better defined in the methods – see comment above for P7, L7

P10, L6: it would be useful to include a figure that shows Landsat images of these stable glaciers to prove that they haven't changed, and so that the velocity profiles in Fig. 7 can be understood in relation to conditions on the ground. This could be a supplementary figure if there isn't space in the main text.

P10, L18: remember to include reference to Figure S1 in this section (e.g., for E10, W8).

P11, L3: I have to admit that I was pretty unconvinced that some of these glaciers were surge-type based on the low resolution images provided in Fig. 8. So I went to Google Earth and used the 'historical imagery' time slider to find a large number of excellent, high resolution images that cover your glaciers of interest. Using these, it is much easier to prove that surges occurred – for example, see screen captures on the next page for glacier E12, which show a dramatic change in surface crevassing between 2007 and 2011. Similarly, the changes for glaciers E13 and W13 in Google Earth are much clearer than shown in Fig. 8. The Google Earth images also help to correct what appear to be some misinterpretations in your figures, such as the label for the 'Stable glacier terminus' for glacier E7 in Fig. 8, which appears to be ~2 km north of the actual terminus (I'm also unconvinced whether E7 is actually a surge-type glacier based on Google Earth).

So I believe that you need to go back through the analyses for all 28 glaciers in your study and using the high resolution Google Earth imagery (perhaps also Bing Aerial: https://www.bing.com/maps) to supplement the Landsat and ASTER data you already use. This would make your interpretations for some glaciers much stronger, while potentially correcting misinterpretations on others. This would also enable a better description of the features indicative of surges, such as changes in crevassing, terminus advance, looped moraines, potholes, etc. (could list these in a table for each glacier).

[Figure]

[Figure]

Glacier E12 from Google Earth: 2007 vs. 2011: I'm now convinced that it surged!

P12, L1: I expect that some of these interpretations might need to be updated after analysis of the Google Earth imagery

P12, L20: please provide an indication of how many (and exactly which) of these glaciers have not previously been identified as surge-type in other databases

P13, L10: change 'velocity initiated' to 'velocity event initiated'

P13, L16: reference to Hewitt's discussion of thermal vs. hydrological causes (and potential role of trunk-tributary interactions) of the surges of nearby Karakoram glaciers would be useful to include here: Hewitt, K., 2007: Tributary glacier surges: an exceptional concentration at Panmah Glacier, Karakoram Himalaya. *Journal of Glaciology*, 53: 181–188.

P14, L6: when you look at the high resolution Google Earth imagery I don't believe that all the termini were unaffected by the surges

P14, L12: what does 'and so on' refer to exactly?

P15, L1: reference to Copland et al. (2011) would seem to be useful here, as they talk about Karakoram surge periodicity in detail:
Copland, L., Sylvestre, T., Bishop, M.P., Shroder, J.F., Seong, Y.B. Owen, L.A., Bush, A. and Kamp, U. 2011. Expanded and recently increased glacier surging in the Karakoram. Arctic, Antarctic, and Alpine Research, 43(4), 503-516.

P15, L21: I think that it would be more realistic to say 'surge events *likely* remain undiscovered'

P16, L5: based on my earlier comment, I'm not convinced that you did see any receding glaciers; I think that your only retreating glacier (W5) is the result of a quiescent phase after a surge.

P17, L9: the statement that 'KM glaciers on the northeastern slopes are also in a period of rapid recession' seems to directly contradict what you just said and what you show in Fig. 2: i.e., that every glacier on the northeastern slopes is either stable or surge-type. With no evidence to back up your statement I think that you should therefore delete it.

P29, Fig. 4: I find the use of the white lines a bit confusing as the same terminus outline from a particular date is solid in the first part of each image, but dashed in the second part. It would be easier to follow if you used a consistent colour over time; e.g., blue for all early outlines, white for all recent outlines. Same comment applies to Figs. 6 & 8.

P32, Fig. 8: indicate what the white and red lines indicate in the caption

P34, Fig. 10: what do the letters a to g indicate?
P36, Table 1: state the year that the values refer to for 'area' and 'debris cover', and the period that the column 'area changed' refers to. And why are area and debris cover values not provided for 3 of the glaciers listed in the table?

Figure S1: this image quality is pretty low – perhaps it has suffered from compression in conversion to a PDF (size of the entire PDF file is only 147 kb). Can you therefore produce this figure at much higher resolution? There is also no need to make the figure so small; the luxury of having a supplementary section is that you have lots of available space, so I would prefer to see the images for each glacier made much larger (e.g., use the full page width for W8, E9, etc.). Also please indicate what the dotted and dashed lines indicate.

---

## Referee Comment (RC2) · Anonymous Referee #2 · 24 Sep 2018

This is the first report on the characteristics of glaciers in the Kingata Mountains. Because of its remote location and poor accessibility, it is natural for the authors to analyze satellite remote sensing imageries. However, the authors analyzed only optical images with limited period and area, which did not lead to significant scientific novelties. The analyzed location is certainly unique, but the presence of surge-type glaciers in High Mountain Asia is no longer surprising. There exist more glaciers with longer length to the southeast of the studied area in this study. Why did the authors limit the analysis area?

In addition, the velocity profiles with nearly one-year temporal resolution will prevent

us from examining the seasonal changes, and the authors' conclusion regarding the surge mechanisms cannot be supported from the present data.

The classification map in Figure 2 and terminus changes in Figure 3 are quite similar to Figures 3 and 4 in Yasuda and Furuya (2015), but they employed both optical and radar images with much longer period since 1970s. The criteria of the authors' Figure 2 are not clearly mentioned and uncertain, either.

Figure 10 is overly qualitative and speculative.

---

## Author Comment (AC1) · 13 Oct 2018

**Response to Reviewer #1**

Dear Anonymous Referee #1,

Thank you very much for your detailed and supportive review.  We particularly appreciate your major suggestion to review other sources of satellite imagery and having done this, now feel that our interpretations have much greater support. Here, we respond (in black plain text) to your comments (in blue italics) one by one. We also attach a marked-up version of revised manuscript and supplement.

Best regards,

Guang Liu, Mingyang Lv and Co-authors

*1. P2, L10: it would also be useful to mention the nearby Karakoram here (which is already described in several of the papers you reference). An added reference to this paper would also be useful:*

*Gardelle et al. 2012. Slight mass gain of Karakoram glaciers in the early twenty-first century. Nature Geoscience, 5, 322-325*

Thanks for this suggestion. Certainly, given that we include the Karakoram in the discussion of our results it would be sensible to also include it here. We have amended the text accordingly and added the Gardelle et al., 2012 reference too.

*2. P2, L13: clarify the period that the reference to positive mass balance of the eastern Pamir refers to*

Yes, this was an omission on our part. We now add 'in the early twenty-first century' right after 'with the eastern Pamir experiencing a slight positive mass balance'.

*3. P4, L14: the distance from Taxkogan meteorological station to the Kingata Mountains is >100 km, so this should be noted.*

We agree, and have added some text to clarify: 'Taxkogan meteorological station (37°46'N, 75°14'E; 3090.9 m asl) is located near to the Muztag Ata mountains, approximately 150 km away from KM.' We have also made some necessary adjustment to the following sentence.

*4. P4, L16: clarify where a 'mean temperature as high as 15°C' is referring to: e.g., over glaciers? At which altitude?*

Thank you for pointing this out. According to Shangguan et al (2006), 15 °C is referring to the temperature at the same altitude of the meteorological station. We have added 'as high as 15 °C (at an elevation of ~3000 m asl)' to the text.

*5. P5, L4: I think that 'high accuracy' is optimistic given the resolution of your imagery; 'good accuracy' would seem to be a better descriptor*

We agree that 'good accuracy' is a better descriptor here and have changed the text accordingly.

We meant to refer here to the common projection, and so have changed 'dataset' to 'system' and added 'specifically to the Universal Transverse Mercator (UTM) grid'.

*7. P6, L10: state which band(s) and resolution were used for the determination of glacial motion; in part, this is needed to understand the pixel size being referred to here*

We have added a sentence here to address this: 'All imagery used for surface displacement detection had a spatial resolution of 15 m (i.e. panchromatic bands of Landsats 7 (ETM+) and 8 (OLI), and ASTER band 3N).

*8. P7, L6: state the averaging distance used for the velocity smoothing, and how blunders were identified in the data*

We have added 'with a kernel size of 3 × 3 (45 m × 45 m in horizontal distance)' to describe the Gaussian Low Pass filtering method we use. Also, we give a clear identification to blunders adding a sentence 'We took values that differed by > 20 m/yr, compared to those in surrounding areas, as blunders'.

*9. P7, L7: how do you know that the accuracy of manual measurements is within 30 m, and therefore which area changes can be considered significant? There are some useful papers that address this issue directly, such as:*

*⬚ Hall, D. K., Baa, K. J., Schöner, W., Bindschadler, R. A., and Chien, J. Y. L., 2003: Consideration of the errors inherent in mapping historical glacier positions in Austria from the ground and space (1893–2001). Remote Sensing of Environment, 86: 566–577.*

*⬚ Paul, F., and 19 others, 2013: On the accuracy of glacier outlines derived from remote-sensing data. Annals of Glaciology, 54(63): 171–182.*

Thanks for highlighting this. We have revisited our error analysis and applied the methods in the suggested papers. Specifically, we selected five differently sized glaciers in our study area and manually digitized their termini outlines both in 1999 and in 2016 for five times independently, and then measured the changes for each glacier along their centreline. We take the average standard deviation of the five glaciers as our manual measurement accuracy. We have revised the text in our manuscript to read 'Manually digitizing glacier outlines may give inconsistent and unreproducible results (Hall et al., 2003). We therefore used the method recommended by Paul et al, (2013) to measure the associated error. This entailed repeatedly digitizing the termini position of five differently sized glaciers in our study area and calculating the standard deviation of changes along their centrelines. The final accuracy of our manual measurement was calculated as 15.7 m. A combination of Landsat multipsectral, Landsat panchromatic and high-resolution images in Google Earth helped us to determine the sign of terminal changes'. We add the two papers you mentioned to the references.

*10. P7, L18: be more explicit as to what 'discrete changes in glacial motion' refers to*

We have added 'such as a clear acceleration or deceleration in some part or along the entire glacier within adjacent years' to the original sentence.

*11. P8, L9: state what the total glacier area was in 1999 so that the 1.33 km2 change can be put in perspective*

Thanks for the suggestion. We have changed the original sentence to now read 'The total glacier area in 2016 was 357.19 km2, an increase of more than 1.33 km2 since 1999'.

*12. P8, L13: can you be more specific about the duration of the individual surges? A range of 'a few months to several years' is pretty broad*

See response to point 13 immediately below.

*13. P8, L14: please better describe what the 'clear sign of surge behaviour' was – e.g., how far did the glacier advance over which period? Fig. S1 is too small and too low resolution to make out much meaningful detail (also see specific comments below). Please improve!*

We discussed this at some length and decided the clearest way to do this would be to add a table describing each event and the evidence we have for interpreting it as a surge (see Table 2 below). We also amended the specific text referred to above from 'it showed clear sign of surge behaviour' to 'there was a clear terminus advance between…'.

**Table 2: Detailed information of each surge-type glacier in this study including evidence of each surge event, their initiating and terminating year, and their duration**

| No. | Evidences of surge events | Surge initiating year | Surge terminating year | Surge duration |
|---|---|---|---|---|
| E1 | Surface features show clear movement; Looped moraines; A clear surge front in velocity profiles. | 1999 | 2003 | 4 years |
|  |  | 2013 | 2016 | 3 years |
| E7 | Surface features show clear movement; Looped moraines; A clear surge front in velocity profiles. | 2013 | After 2016 | > 3 years |
| E9 | Destruction of original surface feature; Changes in ice crevasses; Looped moraines; A clear acceleration near terminus region. | 2007 | After 2016 | > 9 years |
| E10 | Terminus advanced 588 m; A clear acceleration and deceleration along glacier tongue during study period. | 2007 | Before 2010 | < 3 years |
| E11 | Clear surge fronts in satellite images; Changes in crevassing; Looped moraines; Broken surface feature; Disappearance of glacial ponds. | 1999 | 2002 | 3 year |
|  |  | 2013 | 2016 | 3 years |
| E12 | Clear surge front in satellite images; Changes in ice crevasses; Looped moraines; Disappearance of glacial ponds; A clear surge front in velocity profiles. | 2007 | After 2016 | > 9 years |
| E13 | Clear surge front in satellite images; Changes in ice crevasses; Looped moraines; Disappearance of | After 2005 | 2007 | < 2 years |

| | | | | |
|---|---|---|---|---|
| | glacial ponds; A clear deceleration along glacier tongue. | | | |
| W6 | Clear increase and decrease of bare ice area; A clear acceleration near its accumulation zone. | After 1977 | Before 1989 | < 12 years |
| W8 | Terminus advanced 1431 m; A clear deceleration along glacier tongue. | 1993 | 2002 | 9 years |
| W9 | Terminus advanced 810 m; Changes in ice crevasses. | 2007 | 2008 | 1 year |
| W12 | Terminus advanced more 1435 m; Changes in ice crevasses; Looped moraines; Abnormal change in velocity profiles. | 2001 | 2007 | 6 years |
| W13 | East branch advanced ~450 m and squeezed main branch; Changes in ice crevasses; Velocity increased near terminus region of trunk glacier. | 2008 | 2010 | 2 years |
| W5 | No obvious sign of surge except an advance of 161 m between 1972 and 1977; Distorted moraines in lower terminus region; Terminus retreated while surface velocity increased from 1999 to 2016. | Before 1972 | | |

*14. P8, L18: please specify the end period for the movement of '100 m/yr since 2007'. Also specify the period that 'termini of advancing glaciers… changed less than 300 m' refers to: my initial assumption was 1999-2016, but in the next sentence you say that remained stable until 2007*

5  Yes, re-reading this we can see it is somewhat confusing. We have amended the text to read 'the east branch surged into the main branch at a rate of approximately 100 m/yr from 2007 to 2016' and added 'from 1999 to 2016' to specify the period we are referring to for the advancing glaciers.

*15. P9, L6: I don't understand what 'displacement following the entire glacier' means. It would also be useful to state what the change in glacier length was, in addition to area*

10  We have clarified both of these points by amending the text to read 'but they all advanced (W2: 245 m, W3: 212 m, W10: 154 m, W11: 224 m) leading to a total increase in area of 0.38 km$^2$'.

*16. P9, L13: change 'truck glacier' to 'trunk glacier'!*

Thanks for picking this up! We've changed it accordingly.

*17. P9, L15: to me, W5 looks like a surge-type glacier due to the distorted moraines in lower terminus region in*
15  *2000 (in particular it looks as if the tributary to the SE might have recently surged; see Google Earth for high resolution imagery). It would be worthwhile looking through old Landsat imagery to see if this surge was captured, which would prove this definitively. I would therefore argue that the unusual recession of this glacier is primarily due to the quiescent phase of the surge cycle, which would make sense when no other glacier in the KM retreated over the study period.*

We have discussed this glacier at some length following your comment, and concluded that given its surface morphology and measured surface velocities, it is indeed most likely to be a surge-type glacier in quiescence. As suggested, we have reviewed all available Landsat images back to 1972. The only indication of a possible previous surge is between 1972 and 1977, during which time its terminus advanced 161 m. From 1977 to 2016, we do not find other obvious sign of surge events. We have also checked high resolution images in Google Earth and can confirm that the tributary to southeast did not surge recently. Nevertheless, given the distorted moraines in the lower terminus region and the terminus and surface dynamics measured between 1999 and 2016, all of which indicate a surge-type glacier in its quiescent phase, we take W5 as a surge-type glacier. We have changed all related parts in manuscript including figures to reflect this adjustment in interpretation.

*18. P10, L2: you mean that these glaciers showed no detectable change beyond error limits? This is why the errors need to be better defined in the methods – see comment above for P7, L7*

Yes, we do mean within the error limits, and have amended the text to read 'Eleven glaciers in our study area are classified as being stable over the study period having shown no detectable change in terminus position or surface features beyond error limits'.

*19. P10, L6: it would be useful to include a figure that shows Landsat images of these stable glaciers to prove that they haven't changed, and so that the velocity profiles in Fig. 7 can be understood in relation to conditions on the ground. This could be a supplementary figure if there isn't space in the main text.*

We agree, and have made a new figure (S1) showing the stable glaciers in KM from 1999 to 2016.

*20. P10, L18: remember to include reference to Figure S1 in this section (e.g., for E10, W8).*

Thanks – added.

*21. P11, L3: I have to admit that I was pretty unconvinced that some of these glaciers were surge-type based on the low resolution images provided in Fig. 8. So I went to Google Earth and used the 'historical imagery' time slider to find a large number of excellent, high resolution images that cover your glaciers of interest. Using these, it is much easier to prove that surges occurred – for example, see screen captures on the next page for glacier E12, which show a dramatic change in surface crevassing between 2007 and 2011. Similarly, the changes for glaciers E13 and W13 in Google Earth are much clearer than shown in Fig. 8. The Google Earth images also help to correct what appear to be some misinterpretations in your figures, such as the label for the 'Stable glacier terminus' for glacier E7 in Fig. 8, which appears to be ~2 km north of the actual terminus (I'm also unconvinced whether E7 is actually a surge-type glacier based on Google Earth).*

*So I believe that you need to go back through the analyses for all 28 glaciers in your study and using the high resolution Google Earth imagery (perhaps also Bing Aerial: https://www.bing.com/maps) to supplement the Landsat and ASTER data you already use. This would make your interpretations for some glaciers much stronger, while potentially correcting misinterpretations on others. This would also enable a better description of the features indicative of surges, such as changes in crevassing, terminus advance, looped moraines, potholes, etc. (could list these in a table for each glacier).*

This is a great suggestion, and we have done exactly as suggested. As a point of interest, the use of Google Earth is banned in China, so we have carried out this extra analysis while the lead author was a visiting researcher in the UK. More details on our revisions are included in the response to point 22 immediately below.

*22. P12, L1: I expect that some of these interpretations might need to be updated after analysis of the Google Earth imagery*

We have now replaced some Landsat images in our original figures with those from Google Earth. We have also included a new supplementary file (S3) containing 16 GIFs that show the changes of some advancing and surge-type glaciers between 1999 and 2016. Based on high spatial resolution images in Google Earth, we improve some parts with a better description of surface feature changes and surge evidence in Session 4 Results. We add a sentence 'High spatial resolution images from Google Earth (data provided by DigitalGlobe, NASA, and Landsat/Copernicus) were also used for detailed visual interpretations of surface change.' to the Session 3 Data and Methods.

For E7, we checked the images in Google Earth carefully and still hold the opinion that its terminus region remained stable during our study period. The yellow line and red line give the changes of visible surface features of E7 before and after the surge in Figure 7, not pointing its terminus position. We admit it is hard to tell the exact terminus position for E7. However, we take E7 as a surge-type glacier mainly based on its velocity profiles (Fig. 8c). It shows a clear surge front from 2013 to 2016.

*23. P12, L20: please provide an indication of how many (and exactly which) of these glaciers have not previously been identified as surge-type in other databases*

We checked the geodatabase of 2317 surge-type glaciers described in Sevestre and Benn (2015). Dr. Sevestre sent us the geodatabase in 2016. We also carried out a comprehensive literature search, but we cannot find any records of surges in KM. Very few publications have ever focused on glaciers in KM. We add one sentence 'None of the glaciers detailed here have been identified as surge-type glaciers in previous publications' to clarify this.

*24. P13, L10: change 'velocity initiated' to 'velocity event initiated'*

Done

*25. P13, L16: reference to Hewitt's discussion of thermal vs. hydrological causes (and potential role of trunk-tributary interactions) of the surges of nearby Karakoram glaciers would be useful to include here:*

*Hewitt, K., 2007: Tributary glacier surges: an exceptional concentration at Panmah Glacier, Karakoram Himalaya. Journal of Glaciology, 53: 181–188.*

Thanks, this is a good suggestion. We have amended the text to read 'They are interpreted to be a result of both hydrological and thermal changes (Kamb et al., 1985; Fowler et al., 2001; Hewitt, 2007)'.

*26. P14, L6: when you look at the high resolution Google Earth imagery I don't believe that all the termini were unaffected by the surges*

We agree, and have changed 'The surges of E1 and E7 showed minor surface feature advances' to 'The surges of E1 and E7 showed new crevasses with yearly detectable surface feature advances'.

*27. P14, L12: what does 'and so on' refer to exactly?*

We have amended the text to read 'and their configurations in relation to neighbouring glaciers, for example'.

*28. P15, L1: reference to Copland et al. (2011) would seem to be useful here, as they talk about Karakoram surge periodicity in detail:*

*Copland, L., Sylvestre, T., Bishop, M.P., Shroder, J.F., Seong, Y.B. Owen, L.A., Bush, A. and Kamp, U. 2011. Expanded and recently increased glacier surging in the Karakoram. Arctic, Antarctic, and Alpine Research, 43(4), 503-516.*

We agree to refer to this paper here.

*29. P15, L21: I think that it would be more realistic to say 'surge events likely remain undiscovered'*

We agree and have amended the text.

*30. P16, L5: based on my earlier comment, I'm not convinced that you did see any receding glaciers; I think that your only retreating glacier (W5) is the result of a quiescent phase after a surge.*

As described above we have reviewed the imagery and agree that this glacier is most likely to be in quiescence.

*31. P17, L9: the statement that 'KM glaciers on the northeastern slopes are also in a period of rapid recession' seems to directly contradict what you just said and what you show in Fig. 2: i.e., that every glacier on the northeastern slopes is either stable or surge-type. With no evidence to back up your statement I think that you should therefore delete it.*

This was a hangover from a previous version and we will delete it. Thank you for pointing it out.

*32. P29, Fig. 4: I find the use of the white lines a bit confusing as the same terminus outline from a particular date is solid in the first part of each image, but dashed in the second part. It would be easier to follow if you used a consistent colour over time; e.g., blue for all early outlines, white for all recent outlines. Same comment applies to Figs. 6 & 8.*

We have changed the original dashed lines to yellow solid lines and given related indication in the captions.

*33. P32, Fig. 8: indicate what the white and red lines indicate in the caption*

Done.

*34. P34, Fig. 10: what do the letters a to g indicate?*

We add 'Profile-a to Profile-g indicate different stages during the active and quiescent phases of a surge-type glacier' to the caption.

*35. P36, Table 1: state the year that the values refer to for 'area' and 'debris cover', and the period that the column 'area changed' refers to. And why are area and debris cover values not provided for 3 of the glaciers listed in the table?*

We add 'Areas of glaciers and debris cover are measured based on images in 2016. Values relating to areal changes were measured from 1999 to 2016.' to the caption. E10, E15 and W13 are composite glaciers as recorded in the Second Glacier Inventory Dataset of China (Version 1.0). So we give a total area for these three glaciers. We also change some alignments of Table 1 to make it clearer to readers.

*36. Figure S1: this image quality is pretty low – perhaps it has suffered from compression in conversion to a PDF (size of the entire PDF file is only 147 kb). Can you therefore produce this figure at much higher resolution? There is also no need to make the figure so small; the luxury of having a supplementary section is that you have lots of available space, so I would prefer to see the images for each glacier made much larger (e.g., use the full page width for W8, E9, etc.). Also please indicate what the dotted and dashed lines indicate.*

We have made a new supplementary figure (S2) comprising bigger, and clearer Landsat images and Google Earth images giving the same information of the original Figure S1. We changed the dotted and dashed lines into yellow lines and added the caption with a sentence 'The yellow lines show the outlines of glacier termini in the earlier images and the white lines show the outlines of glacier termini in recent images'.

**Response to Reviewer #2**

Dear Anonymous Referee #2,

Thank you very much for your comments on our work. While we find them interesting, in most cases it was not clear how we might modify the manuscript so the revised version will remain largely unchanged. Nevertheless, here we respond (in black plain text) to your comments (in blue italics) one by one.

Best regards,

Guang Liu, Mingyang Lv and Co-authors

*1. The authors analyzed only optical images with limited period and area, which did not lead to significant scientific novelties. The analyzed location is certainly unique, but the presence of surge-type glaciers in High Mountain Asia is no longer surprising. There exist more glaciers with longer length to the southeast of the studied area in this study. Why did the authors limit the analysis area?*

There are several points here that we would like to address.

The first point relates to the time period of analysis. We chose to study the period over which the Landsat archive holds imagery with sufficient resolution to be able to extract robust measurements of glacier surface velocity as well as terminus fluctuations. This is a period of 17 years (1999 to 2016), and our analysis extends back to the 1970s for some glaciers where their status as a surge- or non-surge type is unclear. Having now

included an analysis of Google Earth imagery, as helpfully suggested by Reviewer #1, we estimate we have reviewed several hundred images in total, and therefore consider this to be comprehensive.

The second point relates to the area of study. The eastern Pamir comprises several discrete mountain ranges, of which the Kingata Mountains are one. There are two other major ranges here, the Kongur Mountains and Muztag Ata. The choice to focus on the Kingata glaciers alone was made partly to keep our analysis manageable in terms of glacier numbers, and partly because the other two ranges host glaciers with different hypsometric, aspect, slope and geomorphological characteristics. By limiting our analysis to a single discrete range we were then able to make clear interpretations about the controls of recession and advance as per the manuscript discussion. This would not have been possible otherwise.

The third point relates to the novelty of our study. Given that, as you recognise, the location is 'certainly unique', there is clear novelty in simply describing the dynamics of this glacier over recent decades. To additionally identify the number of surging glaciers here, their recent activity, their event evolution, and ultimately make some interpretation on their controls, is a major addition to the literature on this poorly studied area, we would suggest.

2. In addition, the velocity profiles with nearly one-year temporal resolution will prevent us from examining the seasonal changes, and the authors' conclusion regarding the surge mechanisms cannot be supported from the present data.

It is true that there is no analysis of seasonal changes (and nowhere did we suggest that this was an aim of our work). We made our interpretations on surge mechanisms based on the length and style of surge evolution, in comparison to other areas of the world that have a greater abundance of both field and remotely sensed observations.

3. The classification map in Figure 2 and terminus changes in Figure 3 are quite similar to Figures 3 and 4 in Yasuda and Furuya (2015), but they employed both optical and radar images with much longer period since 1970s. The criteria of the authors' Figure 2 are not clearly mentioned and uncertain, either.

Figure 2 summarises the findings of our work – these advancing, surge-type and stable glaciers are described and analysed as such within this manuscript. The only similarity with Figure 3 in Yasuda and Furuya (2015) is that we chose to colour the glaciers as per our interpretation. We agree that our Figure 3 and Figure 4 of Yasuda and Furuya (2015) are presented in a similar way; their style of data presentation here is very clear and this is what we wished to emulate. We will seek advice from the journal's editorial team as to whether they see this being a problem for the revised manuscript.

4. Figure 10 is overly qualitative and speculative.

We agree that this is a qualitative interpretation of surge evolution – this is exactly the nature of conceptual models. We do not agree that the figure is speculative, since it is based on the observations we present in the manuscript. Such conceptual models are useful as a summary for future research to test and either support or

refute. We hope that other researchers will do exactly this, so that we can progress our understanding of this important glacierised region.

[revised manuscript text omitted]

---

## Author Response (AR3)

We would like to thank both the reviewer and the editor for their detailed and supportive reviews of our resubmitted manuscript. We are very glad that the manuscript now is much improved as a result of attentive reviews and will be published after technical corrections. We have done the recommended corrections given by both the reviewer and the editor and list them below (comments are marked in blue italics). We also attach a marked-up version of revised manuscript and supplement.

**Response to Reviewer 1 - Luke Copland**

*1. P1, L15: you don't just use these image sources any more, as you now use Google Earth as well (which provides images with a better resolution than Landsat or ASTER)*

Thanks for your kindly suggestion. We add Google Earth here to the image sources we use.

*2. P6, L16: 'sign' here doesn't seem to be the correct word to use. Something like 'features' or 'characteristics' would seem to be better*

We agree 'characteristics' is better and have changed the text accordingly.

*3. P7, L23: specify over which period the advances listed here refer to*

We add 'from 1999 to 2016' in the text to specify the period this time.

*4. P13, L2: wording at the start of this line is a bit awkward. Suggest changing it to something like: 'hydrological station suggest that this type…'*

Thank you for pointing this out. We change the original text to 'hydrological station suggests that'.

*5. P13, L14: change to 'availability of high spatial resolution…'*

Done.

*6. P15, L15: change 'acknowledged of stable state in mass balance' to 'acknowledged to have a stable mass balance'*

This is also suggested by the editor. We have changed 'acknowledged of stable state in mass balance' to 'acknowledged to have a stable mass balance'.

*7. Fig. 7: the red lines are pretty difficult to see in some parts of this image, such as glacier E7*

Thank you for pointing out this. But we cannot find a proper colour to make the lines in all images easy to see. This time we use blue lines for E7 to show the positions of the same surface features in recent images. For the other glaciers, we still use red lines. We also change the original caption to 'the red lines (blue lines for E7) show…'.

*8. Table 2: this new table is very useful! It would be even better if you could provide more information about the year(s) being referred to in the column 'Evidences of surge events' (which should be written 'Evidence of surge events'). E.g., which year(s) does the statement 'A clear surge front in velocity profiles' for E1 refer to?*

Thanks for this suggestion. We change the original text to 'Evidence of surge events'. We also provide the year being referred to these evidences. We present the new Table 2 below.

| No. | Evidence of surge events | Surge initiating year | Surge terminating year | Surge duration |
|---|---|---|---|---|
| E1 | Surface features show clear movement; Looped moraines; Clear surge fronts in velocity profiles from 1999 to 2003 and from 2013 to 2016. | 1999

2013 | 2003

2016 | 4 years

3 years |
| E7 | Surface features show clear movement after June 2013; Looped moraines; A clear surge front in velocity profiles from 2013 to 2016. | 2013 | After 2016 | > 3 years |
| E9 | Destruction of original surface feature after 2007; Changes in ice crevasses; Looped moraines; A clear acceleration near terminus region since 2013. | 2007 | After 2016 | > 9 years |
| E10 | Terminus advanced 588 m since 2007; A clear acceleration and deceleration along glacier tongue during study period. | 2007 | Before 2010 | < 3 years |
| E11 | Clear surge fronts in satellite images acquired in November 2007 and July 2016; Changes in crevassing; Looped moraines; Broken surface feature; Disappearance of glacial ponds. | 1999

2013 | 2002

2016 | 3 year

3 years |
| E12 | A clear surge front in satellite image acquired in October 2008; Changes in ice crevasses; Looped moraines; Disappearance of glacial ponds; A clear surge front in velocity profiles from 2007 to 2016. | 2007 | After 2016 | > 9 years |
| E13 | A clear surge front in satellite image acquired in May 2007; Changes in ice crevasses; Looped moraines; Disappearance of glacial ponds; A clear deceleration along glacier tongue after 2013. | After 2005 | 2007 | < 2 years |
| W6 | Clear increase and decrease of bare ice area from 1977 to 2016; A clear acceleration near its accumulation zone since 2013. | After 1977 | Before 1989 | < 12 years |
| W8 | Terminus advanced 1431 m since 1993; A clear deceleration along glacier tongue since 2013. | 1993 | 2002 | 9 years |
| W9 | Terminus advanced 810 m from 2007 to 2008; Changes in ice crevasses. | 2007 | 2008 | 1 year |
| W12 | Terminus advanced more 1435 m since 2001; Changes in ice crevasses; Looped moraines; Abnormal change in velocity profiles during study period. | 2001 | 2007 | 6 years |

| W13 | East branch advanced ~450 m and squeezed main branch since 2008; Changes in ice crevasses; Velocity increased near terminus region of trunk glacier from 2013 to 2016. | 2008 | 2010 | 2 years |
|------|------|------|------|------|
| W5 | No obvious sign of surge except an advance of 161 m between 1972 and 1977; Distorted moraines in lower terminus region; Terminus retreated while surface velocity increased from 1999 to 2016. | Before 1972 | | |

*9. Table S1: the inclusion of the cloud cover rate for each image in this table seems to be unnecessary, particularly since this number is notoriously unreliable in glaciated terrain (where the cloud-detection algorithm can get confused with bright snow-covered areas).*

We agree to delete the cloud cover rate for each image in Table S1 and have done so.

*10. Fig. S2: the word 'rest' doesn't make sense as used in this figure caption. Delete or change to something like 'remaining'. The use of the yellow and white glacier termini outlines should also be included for more glaciers: e.g., the termini for E9 are currently difficult to distinguish in the sat images due to lack of these lines*

Thank you for this helpful suggestion. We agree 'remaining' is a better descriptor. For E9, the surge did not affect its terminus position, only resulting in surface feature change. As a result, we change the original caption to 'Comparison of the remaining surge-type glaciers before and after surge events. The yellow lines show the positions of glacier termini or surface features in the earlier images and the white lines show the positions of glacier termini or surface features in recent images'. We also add yellow and white lines to indicate the surface features of E9.

*11. Fig. S3: I like these animated gifs!*

We are very glad you like them! Please be aware that we change the sequence of Fig. S1, Fig. S2, and Fig. S3 according to the editor's comments.

**Response to Editor – Arjen Stroeven**

1. P1, L2: We change 'Eastern' to 'eastern'.
2. P1, L23: We change 'KM' to 'Kingata Mountains'.
3. P1, L28: We change the original reference order to 'Oerlemans, 1994; Meier et al., 2007; Burgess et al., 2013; Gardner et al., 2013'.
4. P2, L7: We change the original references to 'Gardelle et al. 2012, 2013; Gardner et al., 2013; Osmonov et al., 2013'.
5. P2, L14: We add '2015' right after 'Quincey et al., 2011' and delete 'Quincey et al., 2015'.
6. P2, L15: We change the reference order to 'Goldstein et al., 1993; Kenyi and Kaufmann, 2003; Gourmelen et al., 2011'.

7. P2, L17: We change 'cloud- and snow-cover' to 'cloud- and snow-covers'.

8. P2, L26: We change the reference order to 'Clarke, 1987; Raymond, 1987; Harrison and Post, 2003'.

9. P2, L29: We delete '(GLOFs)'.

10. P3, L11: We change the text to 'such as regional water resource managers in the city of Kaxgar where glacial meltwater plays an important role in supporting agriculture and livestock to better understand…' to highlight the importance of glacial runoff to local residence.

11. P3, L14 and L24: We add 'mountains in the near southeast' after 'Kongur Mountains and Muztag Ata' to make the relation clear. We also indicate Kongur and Muztag on the inset map of Fig. 1.

12. P3, L18: We add ', respectively' accordingly.

13. P3, L19: We indicate Oytag Glacier Park on Fig. 1.

14. P3, L23: We change 'and daily water use' to 'and support daily life'.

15. P3, L28: We change the original text to 'the Mediterranean-, Black-, and Caspian seas'.

16. P4, L3: We change the original text to 'Glaciers on its southern slopes are generally smaller than those on its northern slopes, and most glaciers on the northern side are heavily debris-covered, which is likely to impact their dynamic evolution and long-term ablation rates'. We also replace 3 additional uses of 'as well as' with proper prepositions.

17. P4, L5: We delete unnecessary parts and change to original sentence to 'Many glaciers in this region are small (less than 2 km2); here we focus on 28 glaciers with areas larger than 5 km2 and remote sensing observations can be made with good accuracy'.

18. P4, L8 and L9: We change the unnecessary capital letters to small letters.

19. P4, L12: We change 'cover' to 'covers'.

20. P4, L21 and P5, L5: We change the position of the first time defining UTM.

21. P5, L7: We change the reference order to 'Avouac et al., 2006; Ayoub et al., 2009; Herman et al., 2011'.

22. P5, L19: We add 'of China (Guo et al., 2014)' to 'the Second Glacier Inventory Dataset' and delete 'main'.

23. P6, L1: We add 'a' before 'Gaussian Low Path'.

24. P6, L5: We change the unnecessary capital letters to small letters.

25. P6, L7: We delete '(Version 1.0)'.

26. P6, L12: We change 'Paul et al,' to 'Paul et al.'.

27. P6, L20: We change to original text to 'in some part of, or along the entire, glacier within adjacent years'.

28. P6, L26: We delete 'other stable' from text.

29. P6, L27: We move 'E10, E15 and W13 are composite glaciers as recorded in the Second Glacier Inventory Dataset of China' to the Table 1 header and the header now is 'Attributes of the glaciers in this study. Glacier area and percentage debris cover are measured based on images in 2016. Values relating to areal changes were measured from 1999 to 2016. E10, E15 and W13 are composite glaciers as recorded in the Second Glacier Inventory Dataset of China'.

30. P7, L9: We rename S2 to S1 and change related parts in the text and supplement.

31. P7, L14: We change 'less than' to 'of at most'.

32. P7, L20: We change the unnecessary capital letters to small letters.

33. P7, L21: We change 'less' to 'lower percentage'.

34. P7, L24: We add 'and' before 'W11' and delete 'in area'.

35. P7, L26: We rename S3 to S2 and change related parts in the text and supplement.

36. P7, L28: We change 'Fig. 5a, b, and c' to 'Fig. 5a-c'.

37. P8, L11: We change the unnecessary capital letters to small letters.
38. P8, L14: We change the original text to 'covered by heavy debris ranging from 11.8% to 52.1%' and change 'From 1999 to 2016' to 'between 1999 and 2016' in the following sentence.
39. P8, L15: We change 'Fig.' to 'Figure'.
40. P8, L17: We change 'Fig. 6a, b, and g' to 'Fig. 6a, b, g'. We also make the same corrections to P8, L19.
41. P8, L18: We change the original text to 'decrease of 20 – 40 m/yr'.
42. P8, L22: We change the unnecessary capital letters to small letters.
43. P8, L24: We add 'position' after 'position'.
44. P8, L25: We change the original text to 'Five glacier surges have resulted in a terminus advance (E10, W8, W9, W12, and W13) that happened only once between 1999 and 2016. According to the glacier termini evolution in Figure 3 and satellite images in Figures 7 and S1, glaciers E10, W9, and W12…'.
45. P9, L2: We delete the repetitive '.'.
46. P9, L5: Change 'trunk branch of W13' to 'trunk glacier'.
47. P9, L13: We change the original text to 'The satellite images of E11, E12, and E13 clearly show that new surge fronts propagated down-glacier breaking down original surface features and transporting mixtures of ice and debris. These mixtures created new alluvial fans, ice crevasses, and looped moraines, which in addition changed the original glacial outflow (Figs. 7, S1, S2)'.
48. P9, L17: Change 'But' to 'However,'.
49. P9, L28: Change 'Fig. 8a' to 'Figure 8a'.
50. P10, L3: Change 'between 3 and 4 years' to '3 to 4 years'.
51. P10, L9: Add 'the' before 'terminus'.
52. P10, L22: Add 'between 2000 and 2016' after '300 m up-valley'.
53. P10, L23: Delete 'in September 2000'.
54. P10, L24: Add 'a' before 'previous surge event'.
55. P11, L2: We change the unnecessary capital letters to small letters.
56. P11, L16: Add 'the' before 'ablation zone'.
57. P11, L25: Add 'directions' after 'downglacier'.
58. P11, L28: Add ',' after 'In these cases'.
59. P12, L3: Add 'to reach the terminus' after 'these events were inhibited'.
60. P12, L10: We change the original sentence to 'the size of the accumulation area, the width of the outlet, and their configurations in relation to neighbouring glaciers'.
61. P12, L20: Change 'suggest' to 'indicate'.
62. P12, L28: Change 'extreme' to 'extremely'.
63. P12, L30: We change the original text to 'The surge of Kelayayilake Glacier, located to the south of KM, was initiated in the winter months and terminated in the summer months of 2015'.
64. P13, L16: We change the unnecessary capital letters to small letters.
65. P13, L24: Add 'the' before 'Bhutan Himalayan main ridge'.
66. P14, L2: Change 'upglacier' to 'accumulation'.
67. P14, L9: Change 'Fig. 11 b and c' to 'Fig. 11 b, c'.
68. P14, L28: We add 'the' before 'Kelayayilake Glacier'.
69. P15, L8: We change 'KM' to 'Kingata Mountains'.
70. P15, L9 and L14: We delete 'in the KM'.

71. P15, L16: We change 'HMA' to 'High Mountain Asia'.

72. P16, L6: We delete the repetitive sentences.

73. P17, L8; We correct the spell of the author's name.

74. P21, L15: We correct the spell of the author's name.

75. P22, Figure 1: We correct the figure and caption accordingly.

76. P23, Figure 2: We correct the figure accordingly.

77. P25, Figure 4: We change 'the' to 'an' in the caption.

78. P28, Figure 7: We change the last sentence in the caption to 'E11 panels show the panchromatic band of Landsat, except for the middle panel which shows ASTER true colour combination in 2007'.

79. P29, Figure 8: We reference e and g to 'accumulation zone (cf. Figs. 8e, g, i)' on P11, L14.

80. P30, Figure 9: add 'position' after 'its terminus' in the caption.

81. P30, Figure 10: We correct the figure accordingly.

[revised manuscript text omitted]